# Infrared and Visible Image Fusion: Methods, Datasets, Applications, and Prospects

**Yongyu Luo [1] and Zhongqiang Luo [1,2,*]**

1    School of Automation and Information Engineering, Sichuan University of Science and Engineering, Yibin 644000, China; 322085404315@stu.suse.edu.cn
2    Artificial Intelligence Key Laboratory of Sichuan Province, Sichuan University of Science and Engineering, Yibin 644000, China
*    Correspondence: luozhongqiang@suse.edu.cn

**Abstract:** Infrared and visible light image fusion combines infrared and visible light images by extracting the main information from each image and fusing it together to provide a more comprehensive image with more features from the two photos. Infrared and visible image fusion has gained popularity in recent years and is increasingly being employed in sectors such as target recognition and tracking, night vision, scene segmentation, and others. In order to provide a concise overview of infrared and visible picture fusion, this paper first explores its historical context before outlining current domestic and international research efforts.   Then, conventional approaches for infrared and visible picture fusion, such as the multi-scale decomposition method and the sparse representation method, are thoroughly introduced. The advancement of deep learning in recent years has greatly aided the field of picture fusion. The outcomes of the fusion have a wide range of potential applications due to the neural networks' strong feature extraction and reconstruction skills. As a result, this research also evaluates deep learning techniques.  After that, some common objective evaluation indexes are provided, and the performance evaluation of infrared and visible image fusion is introduced. The common datasets in the areas of infrared and visible image fusion are also sorted out at the same time. Datasets play a significant role in the advancement of infrared and visible image fusion and are an essential component of infrared and visible image fusion testing. The application of infrared and visible image fusion in many domains is then simply studied with practical examples, particularly in developing fields, used to show its application. Finally, the prospect of the current infrared and visible image fusion field is presented, and the full text is summarized.

**Keywords:** infrared and visible image fusion; image fusion; multi-scale decomposition; compressed sensing; sparse representation; deep learning; performance evaluation

## 1. Introduction

Image fusion tries to create an image rich in numerous aspects and information using a variety of techniques as an enhancement strategy. Combining the images produced by several sensors is the process of image fusion. The continuous advancement of current science and technology has led to the development of picture fusion technology since images with a single piece of information are unable to satisfy people's needs [1]. Infrared and visible image fusion [2], multi-focus image fusion, medical image fusion, remote sensing image fusion, etc. are the main divisions of image fusion.

Infrared and visible picture fusion is the more commonly utilized technology among the four types of image fusion mentioned above. The visible light band, which has great resolution and an exceptionally detailed texture, is the most consistent with the visual field of human eyes and produces images that are quite similar to those seen in people's daily lives. However, it will be significantly disrupted by shielding, the weather, and other things.  The ability to recognize and identify targets in infrared images allows for the

capture of thermal targets even in the most challenging weather situations, such as heavy rain or smoke. On the other hand, low resolution, fuzziness, and poor contrast are further drawbacks of infrared images. In order to improve the visual sensory level of the image, we can combine the advantages of the two when we use a specific fusion method, resulting in a fused image that not only has clear infrared targets in the infrared image but also has more abundant texture details in the visible image [3].

In practical applications, the merging of infrared and visible images can address a variety of issues. For instance, in some situations, the operator must simultaneously monitor a large number of visible and infrared images from the same scene, each of which has its own unique set of requirements. Humans find it very challenging to combine information from visible and infrared images simply by gazing at a variety of them. In some situations with complicated backdrops, infrared images can overcome the constraints of visible images and acquire target information at night or in low-illumination settings, improving the capability of target identification and recognition. The effectiveness of the work process can be substantially increased and convenience is brought about by fusing infrared and visible photographs. At the same time, a wide range of applications for infrared and visible image fusion exist in the fields of night vision, biometric recognition, detection, and tracking [4]. This highlights the importance of infrared and visible image fusion studies.

Image fusion technology was initially proposed in the 1980s, according to literary sources. The use of the Laplacian pyramid approach in binocular picture fusion was first suggested by Burt et al [5]. Later, Adelson et al. [6] created a fusion technique that builds images with increased depth of field using the Laplacian methodology. Toet et al.'s [7] monitoring of infrared and visible image fusion [8] employed a variety of pyramid techniques. Li et al.'s [9] proposal to use the discrete wavelet transform in the area of picture fusion was made in the 1990s. A binary wavelet transform that is orientable was proposed by Koren et al [10]. In order to fuse visible and infrared images, Waxman [11] et al. devised an image fusion technique based on a biological model of color vision [12]. Academician Mu used three-dimensional reconstruction to apply fusion images gathered from various perspectives [13]. A real-time picture fusion demonstration system was developed in the UK in 2002, and it can help pilots find hidden targets on the ground and enhance their visual sensory capacity when helicopters fly at low altitudes [14]. A brand-new technique for photovoltaic array problem diagnostics based on infrared and visible image fusion technology was put out by Wang et al. in 2003 [15]. The blocked abnormal operating area might be found by comparing the visible image's characteristics to those of the infrared image. Multi-sensor fusion was used by Professor Liu to track multiple targets [16]. U.S. engineers created the DVP3000 and DVP4000 systems based on FPGA in 2005, which could combine infrared and visible photos in real-time and be used aboard military ships [17]. The development of visible and infrared imaging seeker technology for surface-to-surface missiles was studied in 2006 by Ni et al. They developed a dual-channel, four-digital signal processor (DSP) image fusion hardware architecture, used fusion technology to recognize military targets in the visible and infrared spectrums, and increased the precision of surface-to-surface missile seeker guidance [18]. The "Tracker" strategic reconnaissance vehicle, which can integrate infrared, visible light, laser, and other information to improve the reconnaissance capacity of the reconnaissance vehicle, was created by the United States, the United Kingdom, and others in 2007 [19]. A method of infrared and visible image fusion based on particle swarm optimization for facial image verification was proposed by Raghavendra et al. in 2011 [20]. The "Resources 1" satellite, created by China, was successfully launched around the turn of the century. It has an infrared spectrum scanner and a visible CCD camera, both of which China separately created, making a big impression in the field of combining infrared and visible light [21]. An infrared and visible image fusion method based on multi-scale edge-preserving decomposition and guided image filtering was proposed by Gan et al. in 2015 [22]. This method combined the benefits of multi-scale decomposition and guided filtering to effectively suppress artifacts while

preserving the details of the source images. In order to address the issues of high maintenance costs and inconvenient management in temperature monitoring in the petroleum industry and other industries, Yuan et al. devised an intelligent monitoring system based on infrared and visible light images in 2016 [23]. In order to convert the image fusion problem into a multi-distribution simultaneous estimation problem in 2020 and produce a fused image with significant contrast and rich texture details, Ma et al. [24] proposed GANMcC, a multi-classification constraint generation adduction network for infrared and visible image fusion. In 2021, Ma et al. [25] proposed STDFusionNet, an infrared and visible image fusion network based on salient target recognition that can achieve salient target detection and critical information fusion in an implicit manner. The preservation of thermal targets and background texture is accomplished by guaranteeing the intensity and gradient consistency in specific regions, while the salient target mask is introduced to guide the network to detect salient areas in the first step of our proposal. To solve the issue of insufficient fusion impact of infrared and visible light images in outdoor environments, Qiu [26] developed an infrared and visible light outdoor image fusion approach based on the convolutional neural network in July 2022. This method involves preprocessing the input infrared image using a rolling-guided filter. The source image is then transformed into a low-frequency coefficient and a high-frequency coefficient, and the low-frequency coefficient and the high-frequency coefficient are fused using a minimum fusion rule and a fusion rule, respectively. In November 2022, Tang [27] et al. proposed a darkness infrared and visible image fusion method (DIVFusion), which can reasonably illuminate the darkness, encourage the aggregation of complementary information, and address issues with low illumination, texture hiding, and color distortion in visible images taken at night. A distributed fusion architecture for infrared and visible images, called RADFNet, was proposed by Feng [28] et al. in January 2023. It is built on residual CNN (RDCNN), edge attention, and multi-scale channel attention. Three channels are used to perform picture fusion, and a distributed fusion framework is used to fully utilize the output of the previous step's fusion. In order to extract the features of infrared and visible pictures, two channels use residual modules. This minimizes the loss of visible image texture information and infrared image target information. Figure 1 shows the approximate chronology of the infrared and visible image fusion domain approaches.

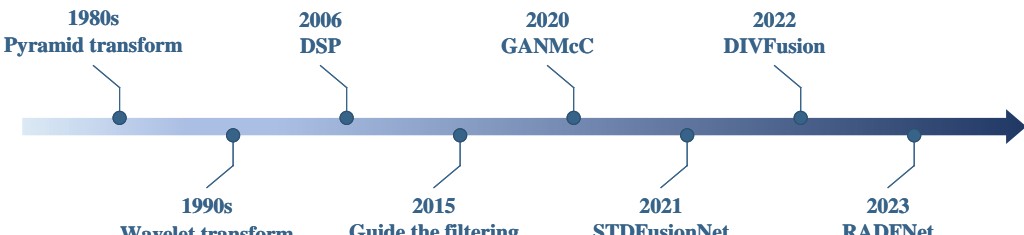

**Figure 1.** Approximate timeline of methods in the field of infrared and visible image fusion.

Figure 2 displays the five-year publication history of works on the fusion of visible and infrared images. All scientific journal papers pertaining to infrared visible image fusion are represented in the bar chart's blue portion, and review articles are represented in the orange portion. With the naked eye, we can observe that from 2019 to 2022, there was an increase in the number of papers published annually on infrared and visible-light-related themes, which amply illustrates the rapid growth of infrared and visible light image fusion. Since there are not many review articles, this study will briefly discuss the infrared and visible image fusion field in an effort to inspire researchers just starting out in their careers.

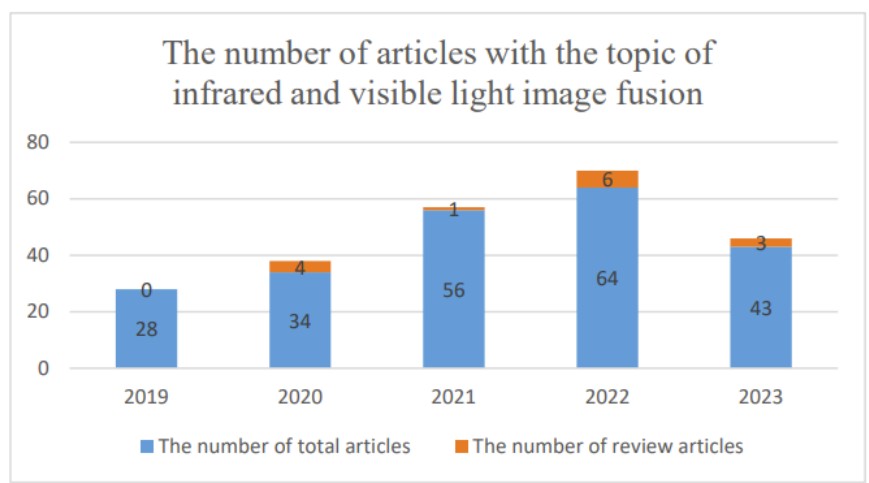

**Figure 2.** The number of articles with the topic of infrared and visible light image fusion (Statistics with Web of Science).

This paper is organized as follows: In the second section, the infrared and visible image fusion techniques are explained in depth, and the benefits and drawbacks of each technique are outlined. The third section lists the fusion image evaluation indexes and explains how these indexes assess the fusion image's quality. The infrared and visible image fusion datasets are sorted in the fourth section, and the distinctions between the various data types are outlined. The fifth section provides a brief overview of infrared and visible image fusion applications. Section 6 describes the problems and potential future prospects for infrared and visible image fusion research. Section 7 concludes by summarizing the thesis. Figure 3 presents a schematic illustration of the article's structure.

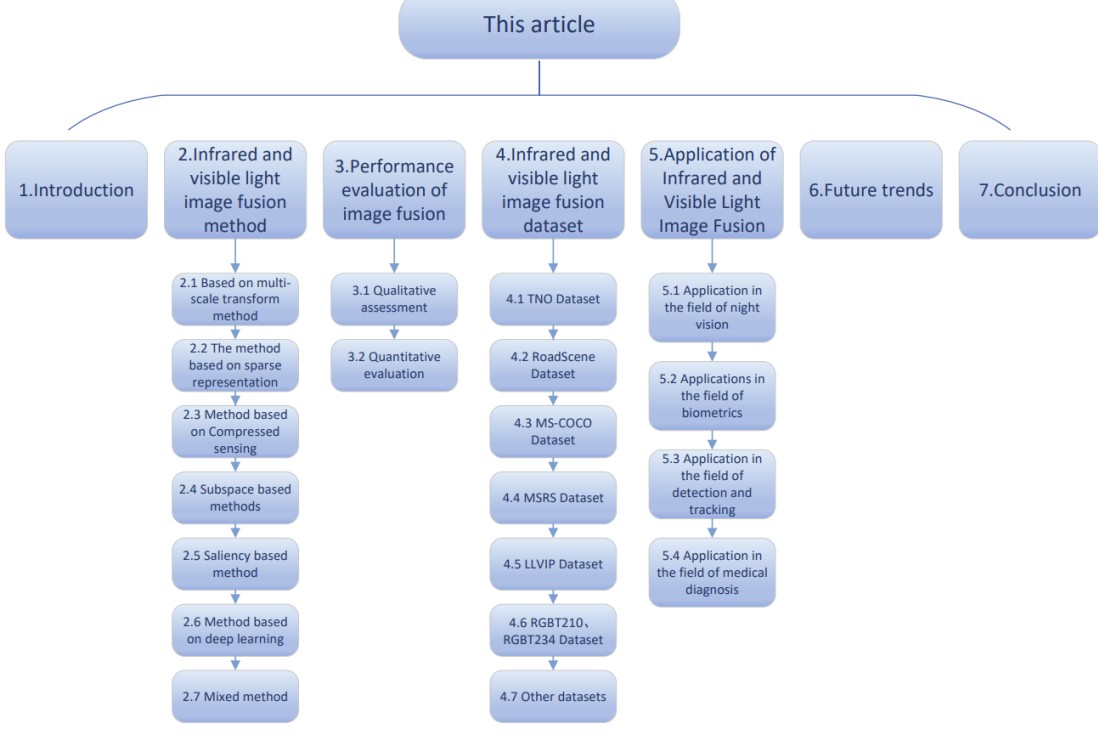

**Figure 3.** Structure of the survey.

## 2. Infrared and Visible Image Fusion Methods

According to the initial research results, image fusion methods can be roughly divided into the following two categories: spatial domain fusion methods and transform domain fusion methods. In spatial domain fusion, we directly process the image's pixels to produce the desired image outcome. In transform domain fusion, the image will first be transferred to the frequency domain, where we need to carry out Fourier transform image processing and then inverse Fourier transform to get the image result we want. The initial image classification techniques can no longer fulfill the requirements due to the advancements in image fusion technology in recent years. According to the existing theoretical technologies, there are several different types of image fusion techniques that can be classified, including multi-scale decomposition, sparse representation, neural networks, sub-space, significance, hybrid, compressed sensing, and deep learning. The infrared and visible picture fusion technique introduced in this section is depicted in Figure 4. The procedures in the figure are briefly described in the paragraphs that follow.

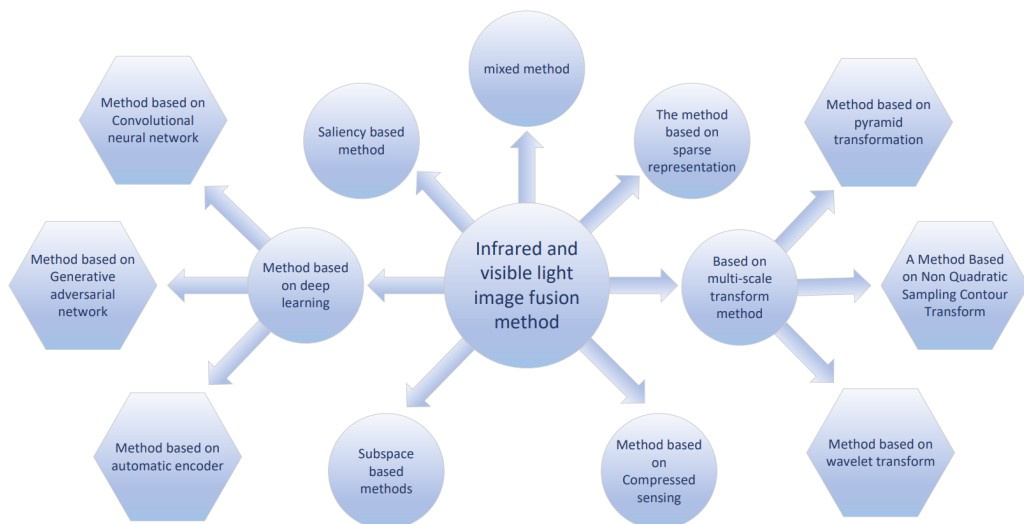

**Figure 4.** Infrared and visible image fusion methods.

### 2.1. Methods Based on Multi-Scale Transformation

Multi-scale transformation has been extensively applied in the field of infrared and visible picture fusion throughout the last few decades. Real-world objects are made up of parts at many scales, and multi-scale transformation can break down the original image into parts at several scales, where each part denotes a sub-image at each scale [29]. The three steps that make up a multi-scale transformation are typically as follows: (1) The source picture is decomposed to produce the high-frequency and low-frequency sub-band layers; (2) Fusion rules are chosen by the characteristics of the various sub-band layers to produce a multi-scale representation; (3) To obtain the fused image, an inverse multi-scale transformation is performed. The choice of transformation and fusion rules, which primarily comprise pyramid transformation, wavelet transformation, non-secondary sampling contour transformation, etc., as illustrated in Figure 5, is the most crucial aspect of multi-scale transformation.

When there is noise or artifacts in the input image during the image fusion process, the quality of the fusion image will be somewhat impacted. For instance, noise and artifacts will impact the image's sharpness and fine details, causing the fused image to lose those details; it may also alter the image's color distribution, causing the fused image to have color distortion. Furthermore, it may result in discontinuous edges in the image, giving the fused image an unsmooth appearance, as well as a reduction in contrast, which would make it difficult to distinguish between light and dark areas in the image. Therefore, among the various picture fusion techniques, the more effective the noise reduction or artifact removal,

the higher the quality of the resulting images. Table 1 lists the benefits and drawbacks of several multi-scale transformation techniques so that we may more easily and intuitively understand how well each technique performs.

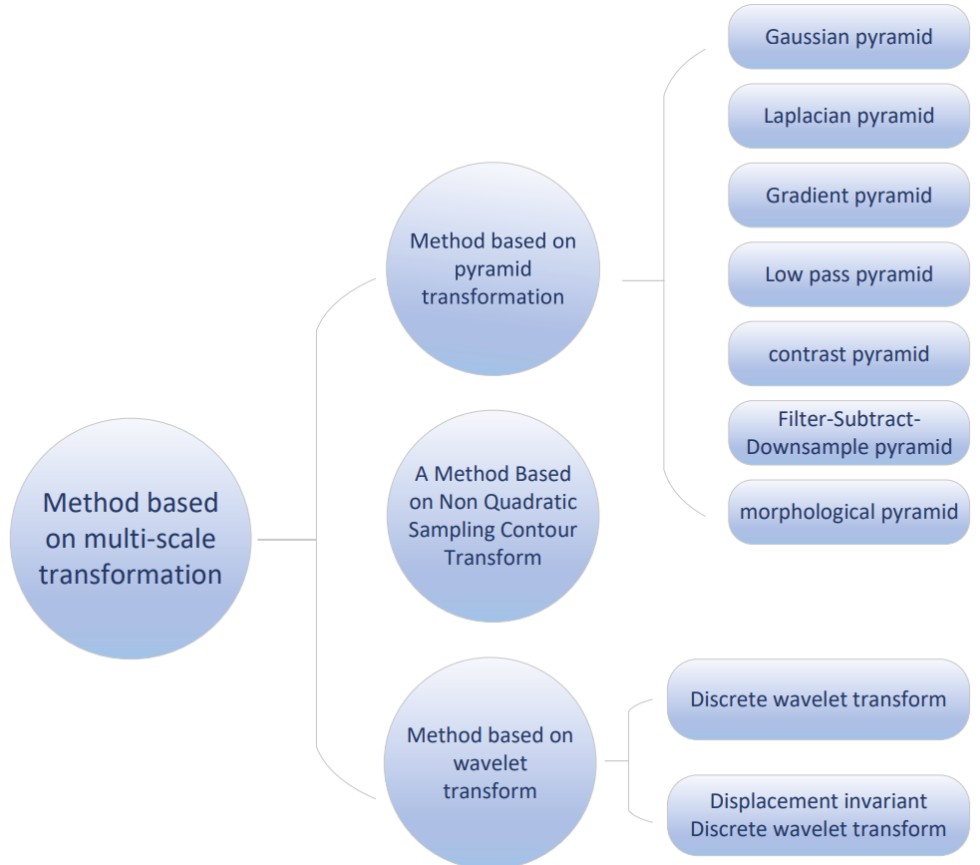

**Figure 5.** Multi-scale transformation methods.

**Table 1.** The advantages and disadvantages of multi-scale transformation methods.

| Multi-Scale Transformation Methods | Typical Methods | Advantages | Disadvantages |
| --- | --- | --- | --- |
| Based on pyramid transformation methods | Gaussian pyramid | Reduced image size and preserved detail | Some information is missing and edges are blurred |
| | Laplacian pyramid | The realization is simple and the operation speed is fast | The fusion result has low contrast |
| | Gradient pyramid | Good anti-noise performance | Poor fusion effect |
| | Low-pass pyramid | The low frequency information of the image is retained, which makes the fusion result more natural | Easy to lose high-frequency details, high computational complexity |
| | Contrast pyramid | The image contrast is improved and more detailed | The anti-noise performance is poor |
| | Filter-subtract extract pyramid | The algorithm is simple and the calculation efficiency is high | Image resolution degradation |
| | Morphological pyramid | Image contrast and brightness are enhanced | Easy to introduce artifacts and distortion |

**Table 1.** *Cont.*

| Multi-Scale Transformation Methods | Typical Methods | Advantages | Disadvantages |
|---|---|---|---|
| Based on wavelet transform methods | Discrete wavelet transform | The anti-noise performance has been improved | The image is prone to aliasing and translation sensitivity |
| | Displacement invariant wavelet transform | Displacement invariance, good retention of high-frequency details | The quality of source image is high and the calculation is complicated |
| Multi-scale geometric analysis | Non-subsampled contour wave transform | Good anti-noise performance and local image processing | Sensitive to image rotation and scaling |
| | Non-subsampled shear wave transform | Poor detail characterization | Translation invariance is good |

2.1.1. Methods Based on Pyramid Transform

- Gauss Pyramid
  A Gaussian pyramid is essentially a multi-scale representation of signals [30], which involves continuously applying Gaussian fuzzy to the same signal or picture and using downward sampling to create numerous sets of signals or images at various scales. The highest picture resolution is found at the base of the Gaussian pyramid, and as image resolution increases, image resolution decreases. The Gaussian pyramid's down-sampling technique is as follows:

  (1) A Gaussian smoothing technique is applied to a given image, i.e., a convolution check image is used for convolution;
  (2) Take a sample of the image, eliminate all even-numbered rows and columns, and create a picture;
  (3) By performing operations (1) and (2) on the image again, the Gaussian pyramid can be obtained [31].

- The Pyramid of Laplace
  Laplacian pyramid [32] fusion is sometimes referred to as multi-band fusion, and its fusion image can be thought of as being formed of information of many frequencies, including a wide variety of features, and the spectrum has a large span. Laplacian pyramid can choose fusion windows of various sizes for various frequency components. The use of a larger fusion window at low frequencies will prevent truncation. Smaller fusion windows will be employed at high frequencies to prevent ghosting. A seamless, ghost-free fusion image is the end product. This is the precise procedure:

  (1) Create a Laplacian pyramid for the two different types of images that need to be combined;
  (2) Create a Gaussian pyramid for the designated fusion region;
  (3) Apply the fusion formula to each layer of the pyramid;
  (4) Use the output image to reconstruct the fused pyramid.

- Gradient pyramid
  The multi-scale decomposition approach based on the Gaussian pyramid is also used in the image fusion algorithm based on gradient pyramid (GP) decomposition [33]. The gradient operator is used for each layer of the Gaussian pyramid image to create the gradient pyramid representation of the image. Each layer's decomposed image comprises information in the horizontal, vertical, diagonal, and other directions of detail, allowing for greater extraction of the image's edge information and much increased stability and noise resistance.

- Low-pass pyramid
  The low-pass pyramid [7] functions by first applying an appropriate smoothing filter to the picture and then twice sampling the resulting smoothed image, that is, twice sampling along each coordinate direction. Repeat the aforementioned procedures for

the generated image numerous times, and after each cycle, a smaller image will be obtained, whose smoothness will grow and whose spatial sampling density, or image resolution, will drop. The smaller images produced by each loop are superimposed on top of the preceding result, creating a pyramid shape if the original image is placed at the bottom. The low-pass pyramid is seen here.

- Contrast pyramid

  The ratio of the contrast pyramid to the low-pass pyramid is close. Contrast itself is defined as the ratio of the difference between the luminance of a location in the image plane and the local background luminance to the local background luminance. The contrast pyramid is a set of images formed by the difference between a series of Gaussian-filtered images and their corresponding low-frequency component images. The amount of information in detailed images increases with pixel size, with the majority of that information corresponding to the edges and area boundaries of the image. Therefore, the pixel value with a high absolute value should be chosen as the pixel value of the fused image on the basis of decomposing the source image into the Laplace pyramid representation. A contrast pyramid-based image fusion technique was introduced by Toet [8] et al. in 1989 in light of the fact that the human visual system is sensitive to local contrast and other visual properties. The contrast1 pyramid is built similarly to the Laplacian pyramid, but each of its layers of images represents the ratio between the two levels that are next to each other in the image of the Gaussian pyramid.

- Filter-Subtract-Extract pyramid (FSD)

  The Laplacian pyramid fusion approach and the filter-subtract-extract method are conceptually equivalent. Their primary distinction is in the procedures used to obtain the many photos needed to build the pyramid. In the Laplace pyramid, the differential image LK of level K is obtained by subtracting from the Gaussian image GK of level K the image that is up-sampled from level K+1 and then low-pass filtered, while in the FSD pyramid, this differential image can be obtained by directly subtracting from the Gaussian image GK of level K the low-pass filtered image of GK. Therefore, the FSD pyramid fusion method is algorithmically more effective than the Laplacian pyramid method since it skips the up-sampling stage.

- Morphological pyramid

  By applying a morphological filter to each layer of the Gaussian pyramid and then determining the difference between the two adjacent levels, one can create the morphological pyramid [7]. Morphological filters are frequently employed to reduce noise and smooth the image, simulating the effects of low-pass filters, but they do not alter the position or shape of the objects in the image. Therefore, fusion using the Laplace pyramid method and fusion using morphological pyramids are essentially equivalent. The only difference is that morphological pyramids are used in place of Laplace pyramids.

### 2.1.2. Based on Wavelet Transform Methods

- Discrete wavelet transform

  The wavelet transform is a multi-resolution image decomposition tool that provides multiple channels to represent the features of an image through different frequency sub-bands at multiple scales [9]. To create the image with the corresponding wavelet coefficient, the source image must first undergo a wavelet transformation. Then, the approximation coefficient and detail coefficient corresponding to the source picture at each level are fused, respectively, in accordance with the fusion principles. A basic weighted average or a weighted average based on PCA (principal component analysis) might be used for this criterion. The final single-output fusion image is reconstructed using the fusion approximation coefficient and detail coefficient of each layer, and this is done using the inverse wavelet transform.

- Displacement invariant discrete wavelet transform

Because the standard discrete wavelet transform (DWT) is not shift-invariant, utilizing it in the fusion approach will provide unstable and flickering results. The fusion output for image sequences must be consistent and steady with the original input sequence. Even a small shift in the source image will cause the DWT output to alter. The Harr DWT approach, which employs a fresh set of filters in each decomposition layer, can be utilized for displacement-invariant fusion in order to preserve DWT translation. This method resolves the issue by skipping the down-sampling stage in the decomposition procedure.

### 2.1.3. Non-subsampled Contourlet Transform

The wavelet transform has been widely applied in various signal processing domains as a quick and efficient way to encode one-dimensional piecewise smooth signals. To separate the discontinuity of edge points, the one-dimensional signal wavelet transform is extended to the two-dimensional image wavelet transform. However, the two-dimensional wavelet transform is unable to capture the rich directional information of the image. Do et al.'s [34] efficient contour transform approach for multi-directional, multi-resolution image representation was offered as a solution to this issue. This model can accurately represent the geometry of picture edges since it is based on the Laplacian pyramid and directional filter banks. However, there are shift variance issues brought on by up- and down-sampling, as well as redundancy issues brought on by the pyramid filter bank topology.

The non-subsampled contourlet transform (NSCT) is a model that has been presented to address the drawbacks of the contour transform [35]. It is quick, flexible, and translation-invariant. The non-subsampling pyramid and non-subsampling direction filter banks are the foundation of the NSCT model. To achieve multi-resolution decomposition, the non-subsampled pyramid filter banks divide each source picture into a collection of high- and low-frequency sub-images. To achieve multi-direction decomposition, the non-subsampled direction filter banks divide these high-frequency sub-images. By doing away with the down-sampling and up-sampling steps of contour transformations, NSCT achieves strong frequency selectivity and regularity and is shift-invariant.

### 2.2. Methods Based on Sparse Representation

The concept of sparse representation (SR) has been extensively used in the disciplines of computer vision [36], machine learning [37], and image processing [38] as a means of describing human visual systems. The goal of sparse representation, in contrast to the traditional multi-scale transformation method, is to learn an overcomplete dictionary from a large number of high-quality natural images with as few information components as possible. The learned dictionary is then used to sparsely represent the source image in order to improve the representation of the image. The intrinsic sparsity of signals is addressed by sparse representation, which imitates the sparse coding techniques utilized by human vision processes. A key component of the source image is thought to be the sparse coefficient. In the field of image processing, the image signal can be expressed as a linear combination of "several" atoms in an overcomplete dictionary. The sparse representation-based infrared and visible picture fusion approach typically consists of four phases [38].

(1) Decompose the source image

Take advantage of the sliding window technique to divide each source image into overlapping chunks. Once the panchromatic image has been down-sampled to the same spatial resolution as the *MS* image, Equation (1) is solved using the least squares method to determine the weight coefficient $g_b$ and the bias constant $\beta_{bias}$. It is possible to determine the final picture $M'_P$ using the linear relationship.

$$I = \sum_{b=1}^{B} g_b M_{MS,b} + \beta_{bias} \tag{1}$$

where $B$ denotes the number of bands and $M_{MS,b}$ denotes the original MS image's $B$-band counterpart. Formula (2) yields the bias constant $\beta_{bias}$ and the weight coefficient $g_b$:

$$M_P^l = \sum_{b=1}^{B} g_b M_{MS,b} + \beta_{bias} \tag{2}$$

(2) Learn a complete dictionary and carry out sparse representation

It is important to acquire the overcomplete dictionary from a large number of high-quality natural images, after which each image block must be sparsely encoded in order to obtain the sparse representation coefficient. Resampling the *MS* image to match the $M_{MS}^l$ panchromatic image in size A window of size $\sqrt{n} \times \sqrt{n}$ is used to traverse each band of the MS image and full-color image from left to right and from top to bottom. Each image block is converted into a column vector of length $n$, denoted as $\left\{ x^{M_{MS,b}^l} \right\}_{i=1}^{N}, \left\{ x_i^{M_P'} \right\}_{i=1}^{N}$, where $N$ is the number of image blocks in a single image and $M_{MS,b}^l$ is the B-band of mass spectrometry images $M_{MS}^l$, $M_{MS,b}^l$, $M_P'$, which is solved by some specific equations.

Low-resolution brightness image's sparse representation coefficient is:

$$\alpha_{I_0} = \sum_{b=1}^{B} g_b \alpha_{M_{MS,b}^l} + \beta_{bias} \tag{3}$$

(3) The sparse representation coefficients are fused according to the fusion rules

The absolute maximum fusion rule partially replaces the panchromatic image's sparse representation coefficient, yielding the high-resolution brightness component's sparse representation coefficient:

$$\alpha_I(i) = \begin{cases} \alpha_{M_P}(i) & |\alpha_{M_P}(i)| > |\alpha_{I_0}(i)| \\ \alpha_{I_0}(i) & other \end{cases} \tag{4}$$

where $i$ is the $i$-th element of the sparse representation coefficient and $\alpha_I$ is the sparse representation coefficient corresponding to the high-resolution brightness component. The sparse representation coefficient that corresponds to the high-resolution *MS* image is found to be:

$$\alpha_{M_{MS,b}}^h = \alpha_{M_{MS,b}^l} + w_b(\alpha_I - \alpha_{I_0}) \tag{5}$$

where $M_{MS,b}$ is the b band of the original low spatial resolution *MS* image, $w_b$ is the weight coefficient corresponding to the b band, defined as $w_b = \frac{\text{cov}(I, M_{MS,b})}{\text{var}(I)}$, and $I$ is the brightness component retrieved in step 1.

(4) Reconstruct fused images

The sparse representation coefficients that result can be reconstructed into high-resolution *MS* pictures using $x = D_h \alpha$ . Figure 6 depicts the flow chart of the sparse representation-based picture fusion algorithm.

Although the sparse representation-based image fusion approach can address the issues of low feature information and high registration needs in multi-scale transformation, it is not without drawbacks. Overly comprehensive dictionaries have restricted signal representation capabilities, which results in the loss of texture information in images. The Max-L1 fusion rule lowers the fused image's signal-to-noise ratio and makes it susceptible to random noise. Little blocks overlap in the slip-window technique, which lowers the algorithm's efficiency.

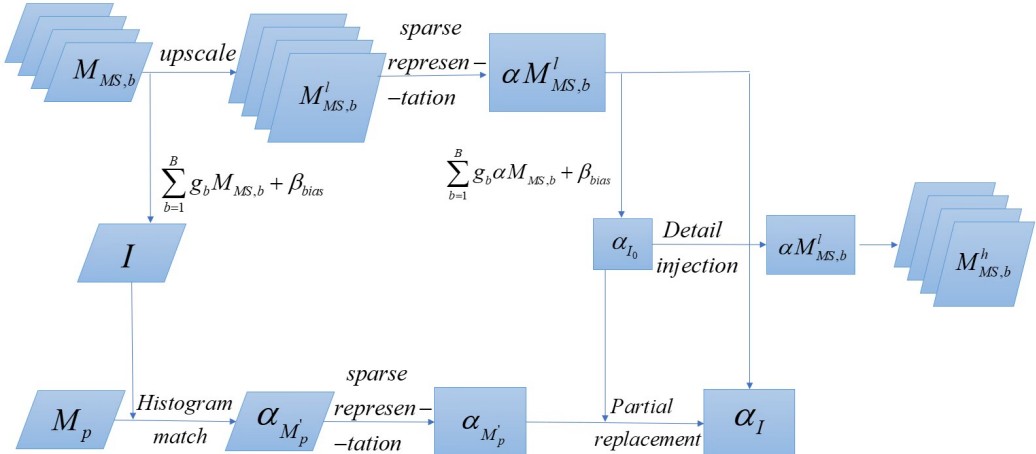

**Figure 6.** Flow chart of image fusion algorithm based on sparse representation.

*2.3. Method Based on Compressed Sensing*

By using a low-signal-sample compression technique based on the signal's sparsity under specific transformations, compressed sensing (CS) can effectively lower the computing complexity and increase the operation rate of the image processing scheme. The sparsity of the signal is the foundation of compressed sensing. The three primary components of the compressed sensing theory are the reconstruction method, coding measurement, and signal sparse representation [39].

The obtained transformation vectors are said to be sparse because they are projected onto the orthogonal transformation basis, and the absolute values of the majority of the transformation coefficients are often tiny. Let $x$ be an $N$-length discrete real-valued signal that can be represented as a concatenation of orthogonal bases. In the time domain, element x is $x(n), n \in [j, 2, ..., N]$. $\Psi^T = [\psi_1, \psi_2, ..., \psi_m, ..., \psi_M]$ is the orthogonal basis, and $\psi_m$ is the $N * 1$ column vector in the expression. A linear combination of orthogonal bases can be used to express $x$:

$$x = \sum_{k=1}^{N} \psi_k \alpha_k = \Psi \alpha \tag{6}$$

In this situation, $\Psi^T$ is the transpose of $\Psi$ , $x\alpha_k = <x, \psi_k> = <\psi_i^T, x>$, x, the N*1 matrix, and $\Psi$, the N*N matrix. The signal $x$ is represented equivalently by. The signal $x$ is referred to as sparse of order $K$ if it has only $K \ll N$ non-zero coefficients on the orthogonal basis $\Psi$. $K$ denotes the signal's sparsity, and $\Psi$ is the signal's sparse basis. The sparsity of a compressible signal is the number of nonzero sparsity coefficients $\alpha K$ with which the signal is represented in the corresponding orthogonal basis $\Psi$ . The original signal is reconstructed from the sparse transform domain using the inverse transform $x = \Psi^H \alpha$ because $\Psi$ is an orthogonal basis and there are hence $\Psi \Psi^H = \Psi^H \Psi = I$ and $\Psi \in C^{N \times N}$.

The sparse signal x is not directly measured in the CS coding measurement model; instead, the sparse signal x is linearly mapped using the measurement matrix $\Phi = [|\varphi_1|, |\varphi_2|, ..., |\varphi_\mu|, ..., |\varphi_M|]$, yielding a set of random measurement values $y_m = <x, \varphi_m>$ in the form of a matrix:

$$y = \Phi x \tag{7}$$

where $\Phi$ is the measurement matrix for $M * N$ , and y is the linear projection of $x$ under $M * 1$'s measurement matrix. When you add it to Formula (7), you obtain:

$$y = \Phi x = \Phi \Psi \alpha = A \alpha \tag{8}$$

where $A = \Phi\Psi$ is the matrix, also known as the perception matrix. Equation (7) converts the N-dimensional signal $x$ to the M-dimensional observed signal y. The dimension of y is substantially lower than the dimension of $x$, and the reconstruction of $\alpha$ from y has an infinite number of sets of solutions, a pathological problem. The initial signal in the Formula (8) is K-order sparse, which means it has only K non-zero coefficients and $K < M \ll N$.

When the original signal's sparse representation coefficient $\alpha$ is known, the original signal $A$ of Equation (8) is solved inversely, yielding the reconstructed original signal. Solving the nonetheless definite equations by the convex optimization method solves the required inverse problem and yields the sparse coefficients $\alpha$ , thus obtaining the original signal X. The measurement number M (that is, the dimension of y) must meet $M = O(K \lg(N))$ , and the measurement matrix $\Phi$ in the measurement Formula (8) must meet the restricted isometric RIP (restricted isometry property) to ensure that the original signal is reconstructed without distortion. When representing an excessively sparse signal using compressed sensing, the measurement matrix's $\Phi$ selection must meet the following qualifications for inequality:

$$1 - \varepsilon \leq \frac{\|A\alpha\|^2}{\|\alpha\|^2} \leq 1 + \varepsilon, \varepsilon > 0 \tag{9}$$

The RIP property of the measurement matrix is a necessary and sufficient condition for the accurate reconstruction of the original signal by compressed sensing theory.

The signal reconstruction algorithm must be implemented during the sampling and compression processes to restore the original signal if the compression matrix satisfies the reconstruction criteria. The original signal can be obtained by solving the underdetermined equations when the number of equations is less than the unknown quantity since the sample data are sparse and the dimension is smaller than that of the original signal. A popular reconstruction methodology for signal reconstruction is the convex optimization method. The original signal $x$ can only be successfully recovered from the $M$-dimensional measurement projection signal $y$ if the perception matrix $A$ in Equation (8) satisfies the RIP condition. Firstly, the sparse coefficient $\alpha = \Psi^T x$ is obtained from the inverse problem of the measured value, and the original signal $x$ is obtained by substituting it into Equation (6). The reconstruction signal $\alpha$ is an optimization problem for the following $l_0$ norm solution (8).

$$\dot{\alpha} = \min_{\alpha} \|\alpha\|_{l_0}, s.t. y = \Phi\Psi\alpha \tag{10}$$

where $\|\alpha\|_{l_0}$ indicates $\alpha$ 's zero norm or the number of non-zero elements in it. Because solving $l_0$ is an NP-hard problem, the identical solution can be achieved by swapping $l_1$ minimal norm solving for $l_0$ minimum norm solving.

$$\dot{\alpha} = \min_{\alpha} \|\alpha\|_{l_1}, s.t. y = \Phi\Psi\alpha \tag{11}$$

The compressed sensing matrix $A$ satisfies the K-RIP requirement, and the measurement matrix is frequently chosen as a random Gaussian distribution matrix to obtain the non-correlation of $\Phi$ and $\Psi$ , allowing the original signal to be reconstructed with high precision. $l_1$ optimization is a convex optimization problem with $O(N^3)$ computational complexity. The most frequent algorithms are the basis tracing method, the interior point method, and the gradient projection method. The internal point method produces precise answers, but it is slow to calculate. The gradient projection method is faster than the interior point method, although it is less precise. The lowest $l_1$ norm optimization issue can be turned into an unconstrained optimization problem to improve computational efficiency.

$$\dot{\alpha} = \arg\min_{\alpha} \|A - y\|_{l_1} + \lambda \|\alpha\|_{l_1} \tag{12}$$

A significant advancement in the field of signal processing, compressed sensing will integrate sampling and compression of signals, enable compressed sampling of signals, obtain fewer measurement data, and enable efficient transmission, storage, and process-

ing [40]. However, it is still necessary to improve the fusion effect of compressed sensing. Consideration can be given to designing efficient analog-to-information conversion front-ends to eliminate the traditional analog-to-digital converters for the compressed sampling of images or designing better reconstruction algorithms to improve the reconstruction of images [41].

### 2.4. Sub-Space-Based Methods

The sub-space-based method is a common method in infrared and visible image fusion. The image that is difficult to distinguish and recognize in the original space is extended to the sub-space by learning the proper sub-space. Samples can be mapped into sub-spaces to get better classification results because sub-spaces offer certain benefits that the original space does not. Visual features in zero-sample image classification are usually extracted by neural networks, and semantic features are obtained by manually defined attributes or keywords extracted from text, which leads to different distributions of visual and semantic features. Poor zero sample identification performance is easily caused by a weak knowledge transfer capacity when the mapping between visual space and semantic space is acquired by direct learning. We can realize the alignment of semantic space and visual space by learning the sub-space, which will improve our capacity for knowledge transfer. High-dimensional input images can be projected into low-dimensional spaces or sub-spaces using sub-space-based algorithms. Low-dimensional sub-spaces can help capture the inherent structure of the original image because redundant information is present in the majority of natural images. Processing low-dimensional sub-space data will also take less time and memory than processing high-dimensional input images [38]. As a result, the degree of generalization can be increased by using low-dimensional sub-space representations. The most common sub-space-based techniques are principal component analysis (PCA), independent component analysis (ICA), and non-negative matrix decomposition (NMF).

(1) Principal component analysis (PCA)

The goal of principal component analysis (PCA) is to reduce the dimension while keeping the information of the original data by transforming several variables that may be associated with unrelated variables into the principal components. The pixel values of all source images at each pixel position must be used in the fusion process. Each pixel value must then be given a weighting factor, and the average of the weighted pixel values for fused images at the same pixel position must be calculated. The best weight factor can then be identified using the procedure. PCA can also eliminate unnecessary data and draw attention to similarities and differences. Principal component analysis has been extensively utilized in many different domains in recent years; however, it is particularly susceptible to gross mistakes, which are common in many applications. To solve this issue, Candes et al. [42] presented resilient principal component analysis (RPCA) in 2011. This method divides the matrix into low-rank and sparse components, which solves the aforementioned issue. RPCA can be used in picture fusion to keep important data and eliminate sparse noise. The RPCA approach can also be used to obtain the sparse components of infrared and visible pictures, and the low and high-frequency sub-band coefficients after fusion can be derived using the sparse components.

(2) Independent component analysis (ICA)

Independent component analysis is a technique for automatically identifying potential factors in a given dataset and is an extension of principal component analysis that aims to transform potentially correlated variables into uncorrelated independent variables. The algorithm has also been successfully used to combine infrared and visible images. ICA-based image fusion techniques typically train a set of bases as the fusion images using a number of natural images with similar content [43]. They can fuse with bases with comparable content once they have a set of bases that have already been educated. A region-based ICA fusion technique was put forth by Cvejic et al. [44] that would segment the image into various areas. The ICA coefficients of each region are

then extracted from the preprocessed pictures. The ICA coefficient is then weighted using the Piella fusion metric in accordance with the quality maximization criterion of the fusion image.

(3) Non-negative matrix factorization

NMF is a component-based object representation paradigm that seeks to decompose the original data matrix into the product of two non-negative matrices [45]. The approach uses a component-based object representation paradigm that is in line with how people perceive things. Therefore, NMF is frequently employed in the field of combining infrared and visible images. Traditional NMF requires a lot of time and is ineffective. Kong et al.'s [46] enhanced NMF, which addresses the issue of random initialization based on singular value decomposition and converges more quickly than conventional random initialization NMF, was used to address these drawbacks.

### 2.5. Saliency-Based Methods

Salience is a technique for drawing attention from the bottom up in the human visual system, which is typically drawn to things or pixels that are more significant than their close vicinity. The saliency-based fusion method can preserve the integrity of the salient target area, improving the visual quality of the fused image in accordance with the workings of the human visual system. First, the source image is divided into significance, detail, and basic layers using significance detection and Gaussian smoothing filters. The target in the image is then highlighted using the highlight layer after the weight coefficient is computed using the nonlinear function. Then, the fusion rule based on phase consistency is used to fuse the detail layer, which makes the preservation of the detail layer better than the traditional maximum-absolute fusion rule. A multi-scale fusion approach based on visual significance graphs (VSM) and weighted least squares (WLS) optimization was put forth by Ma et al. [47] Multi-scale decomposition (MSD) was introduced, and a scroll-oriented filter (RGF) and Gaussian filter were used to decompose the input image into a basic layer and a detail layer. Also suggested is an improved welding method based on VSM. To integrate the detail layer, a new WLS optimization approach is suggested. The merged image can contain more visible features and less noise or unimportant infrared details. In order to better retain the crucial details of the infrared and visible images in the fused image, Liu et al. [48] developed the important mapping of source images in the fusion process. First, using the framework of the joint sparse representation model, the global and local significance maps of the source images are obtained. The local significance map and the global significance map are then combined to create a full significance map using a significance detection model that is then provided. In order to implement the fusion process, a weighted fusion algorithm based on an integrated importance graph is proposed. Decomposing the input image into low-frequency and high-frequency coefficients was done first by transformation, according to Budhiraja et al. [49] The significance-based technique is then utilized to fuse the low-frequency coefficient. Finally, the high-frequency coefficient fusion is performed using the max-abs technique, and the calculation time requirements of various transformations are provided.

### 2.6. Methods Based on Deep Learning

Deep learning is used to perform feature extraction, data fusion, and image reconstruction in order to get around the constraints of conventional techniques. The deep-learning-based image fusion technique can utilize the network's capacity for learning and adaptively train and update the model parameters to create an end-to-end input–output mode. Deep learning significantly lessens the impact of human factors on the results of fusion by avoiding activity-level assessment and fusion rule design in comparison to conventional methods. Furthermore, the deep learning technique fully utilizes the capacity of network feature extraction and fully stores the supplementary data of the original image in the fused image, enhancing the fused image's quality. A range of deep-learning-based fusion algorithms for combining infrared and visible images have evolved in recent years

as a result of the growth of deep learning. Fusion methods based on deep learning can be classified into the following groups: convolutional neural network-based methods, generative adversarial network-based methods, and automated encoder-based methods, depending on the various traits and guiding principles of the algorithms. Table 2 displays the benefits and drawbacks of each deep-learning-based technique.

**Table 2.** The advantages and disadvantages of deep-learning-based methods.

| Deep Learning Methods | Typical Methods | Advantages | Disadvantages |
|---|---|---|---|
| Convolutional Neural Network-based approach (CNN) | VGG-19 network | Contains less artificial noise | It is easy to lose useful information during feature extraction |
| | Fusion framework based on ResNet and ZCA | Retain more structure and edge information | More complicated calculation |
| | FusionDN | Get a single model for multiple fusion tasks to avoid forgetting | Large datasets are required for training |
| | IFCNN | With better generalization ability | If the image is not registered, you need to add a separate image alignment module |
| Generative Adversarial Network-based Approach (GAN) | FusionGAN | Compared with the traditional method, the overall effect is better | The contrast is low and the significance of infrared target is low |
| | FLGCFusionGAN | Improves model efficiency | Large amount of training data |
| | AttentionFGAN | More comprehensive spatial information can be captured to enhance foreground target areas and typical features | The performance of peak signal to noise ratio is weak |
| | SSGAN | The low-level and high-level semantic information of images can be considered comprehensively | The loss of segmentation consistency between the fused image and the source image is not established |
| Autoencoder-based approach (AE) | DeepFuse | It can also produce better fusion effect for extreme exposure images | Because only the last layer of the coded network is used for calculation, useful information in the middle layer may be lost |
| | TCPMFNet | The fusion has good anti-noise and significant contrast | Large amount of computation |

### 2.6.1. Methods Based on Convolutional Neural Network

A convolutional neural network, which is widely used in the field of image recognition, is a kind of artificial neural network that has significant advantages in feature extraction and can provide more information compared to traditional artificial feature extraction methods. The convolutional layer, pooling layer, and fully connected layer are the three layers that make up a convolutional neural network. The whole connection layer is categorized and evaluated, and the convolution layer is utilized to find features. The pooling layer is used to maintain the sample constant while allowing training with fewer parameters and ignoring some pointless information. Figure 7 displays the CNN-based image fusion framework.

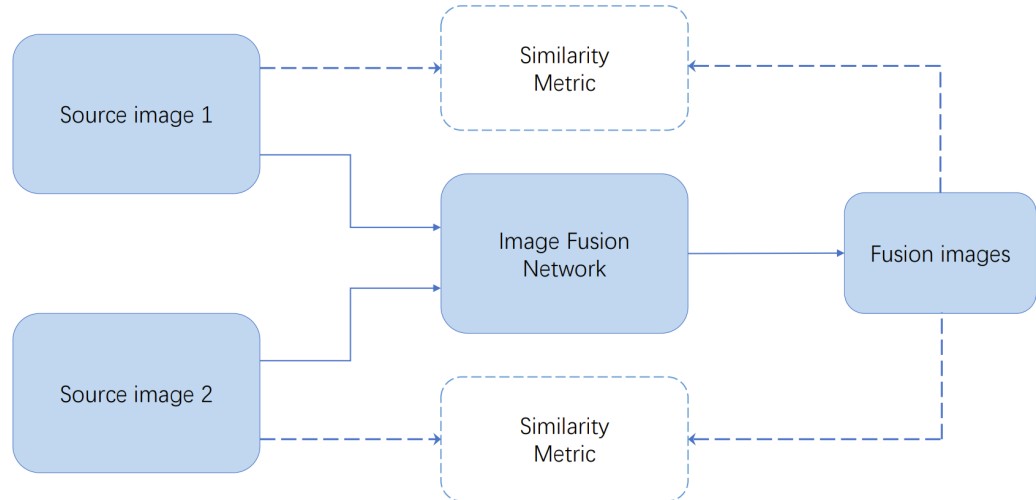

**Figure 7.** Image fusion frame diagram based on CNN.

Under the direction of the loss function, the fusion framework based on the convolutional neural network may achieve hidden feature extraction, aggregation, and picture reconstruction. Convolutional neural networks can also be used to integrate features and provide activity-level measures as a component of a larger fusion framework. A deep learning framework is used by Li et al. [50] to generate all the features of both infrared and visible images as part of their proposed image fusion method. The source image is split into basic and detailed components first. The weighted average approach is then used to combine the constituent pieces. Deep learning networks are used to build multi-scale weighted feature maps and extract multi-layer features for the detailed section. The maximum selection operator then reconstructs the detailed fusion characteristics. Finally, the basic element and the detail part are combined to recreate the fusion image. This technique, however, makes use of the VGG-19 network, and during feature extraction, useful information is lost. Li et al.'s [51] proposal of a fusion framework based on depth characteristics and zero-phase component analysis (ZCA) addressed this issue. First, depth features from the source photos are extracted using ResNet. After obtaining the initial weight map using the local average L1 norm, bicubic interpolation is used to adjust its size to match that of the source image. The weight mapping is then combined with the source picture, and the weighted average technique is utilized to rebuild the fusion image. A new unsupervised and unified dense connected network (FusionDN) is proposed by Xu et al. [52] that is trained to produce fused images with modifications to the source images. At the same time, two data-driven weights obtained by one weight block will retain the feature information of various source images. In addition, rather than creating separate models for every fusion task or roughly combining training tasks, Xu et al. achieved a single model applicable to multiple fusion tasks by using elastic weight merging to prevent forgetting what was learned from earlier tasks when training multiple tasks sequentially. A broad image fusion framework based on a convolutional neural network (IFCNN) was proposed by Zhang et al. [53]. First, features from several input photos were extracted using two convolutional layers. The convolution features are then fused using a suitable fusion rule based on the type of input image. The fused image is then obtained by reconstructing the fused features using two convolution layers.

### 2.6.2. Methods Based on Generating Adversarial Network

The idea of generative adversarial networks (GAN), an artificial intelligence algorithm created to address generative modeling issues, was initially put forth in 2014 by Goodfellow et al. [54] From estimated probability distributions, generative adversarial networks can produce new examples. As soon as it was put forth, it garnered considerable interest in the deep learning community. The generator and discriminator are the two components of

the generative adversarial network. A discriminator is a binary whose inputs are genuine data and sample data generated by the generator. During the training phase, the generator can utilize random noise to make a fresh data sample, but it must provide real photos as much as possible to fool the discriminator. The discriminator makes every effort to distinguish the true picture from the one created by the generator. Finally, after extensive training, you may produce images using the trained generator. Figure 8 depicts the GAN-based framework for picture fusion.

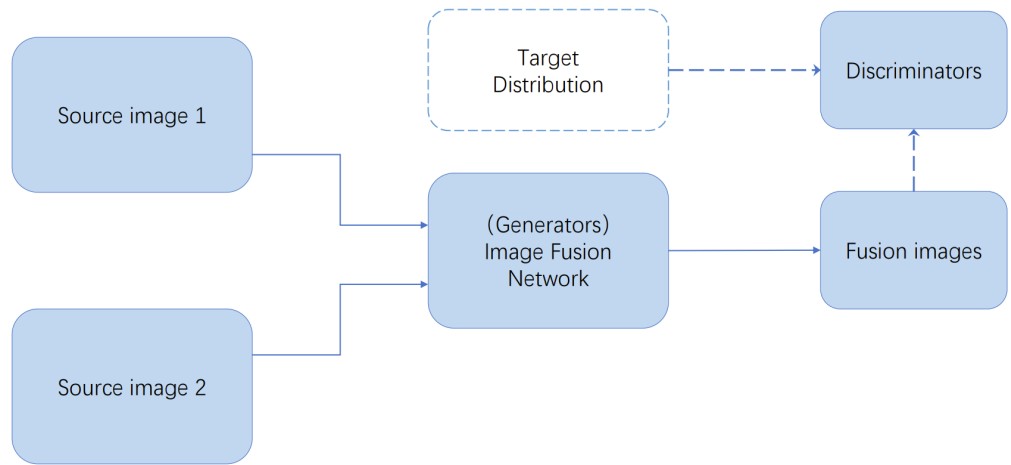

**Figure 8.** Image fusion framework based on GAN network.

Ma et al. [55] used generative adversarial networks for infrared and visible image fusion for the first time, and they proposed a new method to fuse these two types of information, called FusionGAN. The generator primarily creates the fusion image using the additional visible light gradient and main infrared intensity in this manner. The discriminator primarily increases the fused image's clarity and adds more details to the visible portion of the image. FusionGAN is an end-to-end model that forgoes the conventional method of manually creating intricate activity-level measurement and fusion rules in order to prominently display objectives and specifics. However, the convolution procedure consumes too much computer space, and the FusionGAN approach cannot maintain the crucial information of the source picture simultaneously in the fusion process. A brand-new end-to-end network architecture based on generative adversarial networks, known as FLGC-Fusion GAN, was proposed by Yuan et al. [56]. Learnable group convolution in the generator enhances the model's effectiveness, conserves computational resources, and enables us to better balance the model's accuracy and speed. GAN was expanded by Li et al. [51] to include multiple discriminators, and an end-to-end multi-discriminator Wasserstein Generative Adversarial Network (MD-WGAN) was developed. According to this framework, the fused image can keep the texture information provided by the second discriminator as well as the infrared intensity and detail information provided by the first discriminator. To preserve more textures in visible images, a texture loss mechanism based on local binary mode was also designed. An attention-based adversarial network generation method, called AttentionFGAN, was put out by Li et al. [57] for the fusion of visible and infrared images. In order to capture extensive spatial information, the authors incorporate multi-scale attention mechanisms into the generator and discriminator of GAN. This allows the discriminator to concentrate on the background features of the visible image and the foreground target information of the infrared image. An infrared and visible image fusion generative admixture network (SSGAN) based on semantic segmentation was proposed by Hou et al. [58] It can distinguish between the foreground and background of source images through semantic masking and can take into account both low-level features of infrared and visible images and high-level semantic information. This network's generator uses a dual-encoder, single-decoder design that allows it to separate foreground

and background characteristics using various encoder pathways. Additionally, semantic segmentation is used to create the discriminator's input image. The foreground of the infrared image and the background of the visible image are combined to create the image. The thermal target's prominence in the infrared image and the textural features in the visible image can both be preserved in the fused image in this fashion.

### 2.6.3. Methods Based on Autoencoder

In order to transform high-dimensional data into low-dimensional representation, the infrared and visible image fusion approach based on an autoencoder uses a three-layer network of an input layer, a hidden layer, and an output layer. There are typically three steps, which are as follows: The source picture's features are first extracted using an encoder, and then various source image features are combined using a fusion approach. The decoder then reconstructs the fusion image. Figure 9 depicts the autoencoder-based image fusion framework.

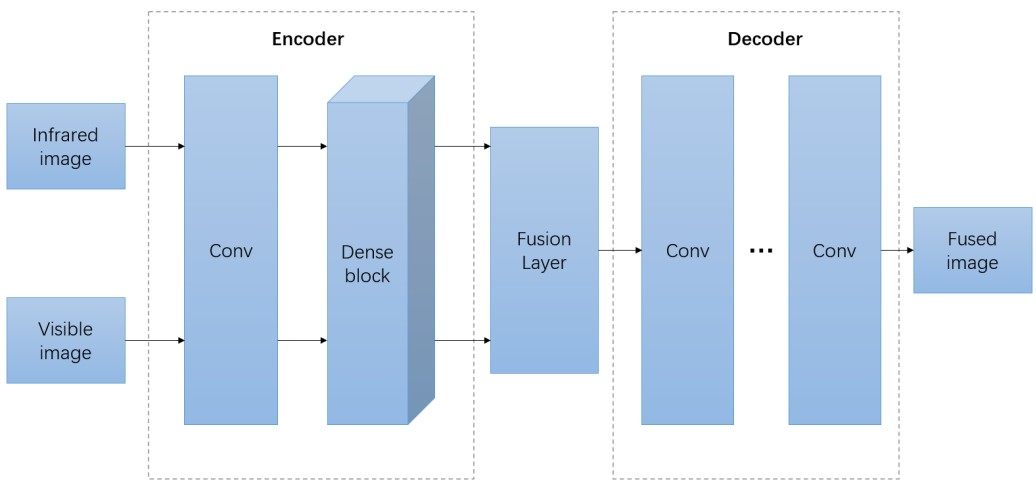

**Figure 9.** Image fusion framework based on autoencoder.

The fusing of multiple exposure photos was made possible in 2017 by Prabhakar et al. [59] who first suggested a unique image fusion architecture based on unsupervised deep learning (DeepFuse). The framework consists of two coding network layers and three decoding network layers; first, convolutional neural networks are used to extract the feature maps from the original image; next, the feature maps in each dimension are fused; and finally, the reduction of the decoding layer is used to produce a fused image. The last layer of the coding network is solely used for calculation in this method, which has a positive outcome but runs the risk of losing useful information in the middle layer. Based on this issue, Li et al. [60] created a complex deep learning network in 2018. Convolutional layers, fusion layers, and dense blocks are combined in the network's coding network, where the output of each layer is coupled to the output of the previous layer. Two fusion layers are created to combine these features, and this structure can extract additional beneficial source image features throughout the coding phase. A double-branch automatic encoder serves as the foundation of Fu et al.'s [61] double-branch network. The authors extract from the encoder the detailed and semantic information of the infrared image and the visible image, respectively. The fusion layer provides the fusion features. The fusion image is then obtained by reconstructing the fusion feature in the decoder. A brand-new self-supervised feature adaptive architecture for infrared and visible image fusion was put forth by Zhao et al. [62]. The original picture information is retained in the first stage using the self-supervised technique in various channels of the coding region; in the second stage, the decoder is replaced with a fusion enhancement module; and finally, a detailed enhanced fusion image is created. An infrared and visible image fusion network (TCPMFNet) using a composite autoencoder and parallel variable-convolution fusion approach was proposed

by Yi et al. [63] The fusion network designs a transformer-convolution parallel hybrid fusion method with outstanding feature fusion performance based on the autoencoder (AE) structure, using the composite autoencoder to encode the rich features of the source picture pairs. Zhang et al. [64] proposed a fusion framework based on self-supervised learning (SSL-WAEIE), designed a weighted autoencoder (WAE) to extract multilevel features that need to be fused in source images, and then further constructed a convolutional information exchange network (CIEN), which can more conveniently complete the fusion contribution estimation of source images.

### 2.7. Hybrid Methods

The aforementioned visible and infrared image fusion techniques each have advantages and drawbacks of their own. The quality of picture fusion will be further enhanced if the benefits of various technologies can be combined. The hybrid approach combines the benefits of various approaches to enhance image fusion performance. Hybrid multi-scale transform and neural network, hybrid multi-scale transform and sparse representation, and hybrid multi-scale transform, sparse representation, and neural network are examples of common hybrid approaches.

In order to improve the target region and preserve details, hybrid multi-scale transform and saliency approaches integrate saliency region detection into a multi-scale transform image fusion framework. A multi-scale decomposition picture fusion technique based on local edge preservation filtering and salience detection was proposed by Zhang et al. [65] The infrared and visible pictures are separated using a local edge-holding filter first. The significant target region in the infrared image is then found using the improved significance detection approach, and the fundamental layer weight of the fusion strategy is established. The fused image is then created by reconstructing each layer. Aiming to address the drawbacks of conventional multi-scale picture fusion techniques, Ma et al. [47] suggested a multi-scale image fusion method based on visual significance maps and weighted least squares optimization. The input image is first divided into basic and detail layers using a rolling guide filter and a Gaussian filter, which introduces multi-scale decomposition. Second, the majority of the fundamental layers produced by multi-scale decomposition will still include low-frequency information in them. An improved method based on a visual saliency map is suggested to fuse the basic layer in order to tackle this issue. To fuse the detail layer, a novel weighted least squares optimization approach is then suggested. The merged image is then obtained.

Sparse representation and neural networks are frequently incorporated into the multi-scale transform infrared and visible image fusion frameworks because they are two of the most frequently used signal and image representation theories in the field of image fusion. A broad image fusion framework was presented by Liu et al. that incorporates multi-scale transform and sparse representation while also addressing the drawbacks of the fusion approach of multi-scale transform and sparse representation. To get the low-pass and high-pass coefficients, a multi-scale transformation is first carried out for each source image that has been previously registered. The high-pass band then uses the absolute value of the coefficient as the activity-level measurement and merges with the low-pass band using a sparse representation-based fusion method. An inverse multi-scale transformation is then applied to the merging coefficient to produce the final fusion image. A visible and infrared image fusion technique based on a non-subsampled shear-wave transformation-spatial frequency-pulse-linked neural network was proposed by Kong et al. [66] NSST not only has advantages in information acquisition and computation savings but also overcomes the disadvantage that shear wave transformation does not have translation invariance. The integration of low-frequency and high-frequency sub-band coefficients into an enhanced IPCNN model is suggested in this research in order to increase function efficiency and protect computing resources. In order to address the issues of poor contrast and ambiguous backdrop features in the existing infrared and visible fusion approaches, Wang et al. [67] suggested a multi-scale fusion method that is based on the combination

of a non-sampled contour wave transform, sparse representation, and a pulse-coupled neural network. The source image is split into a low-frequency and a high-frequency sub-band before being subjected to various scales and directions of NSCT. For the fusion of low-frequency and high-frequency sub-bands, respectively, the fusion rules based on sparse representation and improved PCNN are used. In the end, inverse NSCT is employed to produce the fused image.

## 3. Performance Evaluation of Infrared and Visible Image Fusion

For the quality of image fusion, there are two methods of evaluation: 1. Qualitative assessment, i.e., subjective evaluation method, refers to the visual analysis by a group of observers to match the fused image with the source image, with the visual observation of the human eye as the main body. 2. Quantitative assessment, i.e., objective evaluation method; this refers to the method of quantitative evaluation of the fused image quality by calculating a specific numerical index from the fused image with some specific algorithms. In the following, these two types of methods will be elaborated on in detail [68].

### 3.1. Qualitative Evaluation (Subjective Evaluation Method)

The human visual system serves as the foundation for the subjective evaluation method, which assesses the combined image's quality. Professionals in the field will often receive the fused image for subjective review and provide a professional assessment of the quality of the fused image. The standard technique divides image quality into five levels, with each level's ratings being 5, 4, 3, 2, and 1, respectively, denoting "very good", "good", "average", "poor", and "particularly poor" [69]. A score evaluation of the quality of the fused image will be produced by the expert after scoring the image from their own professional standpoint. The issue with this method is that the results are highly subjective and biased due to the erroneous description. The subjective evaluation approach has the benefits of being logical, uncomplicated, and consistent with the visual features of human eyes, among other things. It must thoroughly examine a number of factors, including spatial details, object size, color brightness, and others. The final evaluation findings will be impacted since many parameters are varied and individual visual variances are significant. A large amount of assessment data from various subjective evaluation individuals is required to improve the accuracy in order to eliminate these uncertainties. This approach, however, is ineffective, time-consuming, and expensive. Therefore, judging the quality of a fusion image solely on the basis of subjective evaluation is unreliable, biased, and ineffective.

### 3.2. Qualitative Evaluation (Objective Evaluation Method)

According to a certain algorithm, objective assessment methods provide quantitative evaluation value and can quantitatively assess the efficacy of fused pictures [70]. The objective evaluation approach has the benefits of being simple, highly effective, highly certain, inexpensive, and highly operable. It has the ability to overcome the limitations of subjective evaluation techniques that are easily influenced by a variety of elements, such as the surroundings. The basic foundations of objective evaluation techniques include information theory, structural similarity, image gradient, statistics, and the human visual system. We will briefly go over a few performance metrics used to gauge the quality of fused images below [71]. Table 3 displays the findings of the examination of the performance metrics for image fusion.

- Peak signal-to-noise ratio (PSNR)
  The peak signal-to-noise ratio [72] primarily evaluates the ratio between the image's effective information and noise, which might indicate image distortion and reflect the quality of the fused image. PSNR is defined as follows:

$$PSNR = 10 \log \frac{X^2}{MSE} \tag{13}$$

In Equation (13), X represents the difference between the maximum and minimum values of the gray scale of the ideal image, which is usually taken as 255. The higher the value of PSNR, the better the quality of the fused image.

- Entropy (EN)

Entropy [73] is mainly an objective evaluation index to evaluate the amount of information contained in an image. If the entropy increases after fusion, it indicates that the fusion information increases and the fusion performance improves. EN is defined as:

$$EN = -\sum_{i=0}^{n} X_i \log_2 X_i \tag{14}$$

In Equation (14), n stands for the grayscale level, $X = \{X_1, X_2, ..., X_n\}$ for the picture's grayscale distribution, and $X_i$ for the normalized histogram of the corresponding grayscale level in the combined image. The more information in the fused image, the larger the EN, and the better the performance of the fusion method is proven. However, noise has an impact on EN, and the more noise, the greater the EN.

- Mean square error (MSE)

The mean square error [74] reflects the difference between the fused image and the reference image and is an objective evaluation index of image quality based on pixel error. MSE is defined as:

$$MSE = \frac{1}{mn} \sum_{i=1}^{m} \sum_{j=1}^{n} (A_{ij} - B_{ij})^2 \tag{15}$$

In Formula (15), $A_{ij}$ and $B_{ij}$ are the gray values of pixels of the fused image and the reference image located in row *i* and column *j*. The smaller the MSE, the smaller the difference between the images, and the better the fusion quality.

- Root mean square error (RMSE)

Root mean square error [75] is an evaluation index reflecting spatial details. RMSE is defined as:

$$RMSE = \sqrt{\frac{1}{mn} \sum_{i=1}^{m} \sum_{j=1}^{n} (A_{ij} - B_{ij})^2} \tag{16}$$

- Nonlinear correlation coefficient (NCC)

The nonlinear correlation coefficient [76] represents the nonlinear correlation between the fused image and the source image. NCC is defined as:

$$NCC = \frac{\sum_{i=2}^{m} \sum_{j=1}^{n} (A_{ij} * B_{ij})}{(A_{ij})^2} \tag{17}$$

- Structural Similarity Index Measure (SSIM)

The structural similarity index [74] is an index used to evaluate the similarity of images A and B, mainly studying the relationship between the change in image structure information and image perception distortion. SSIM is defined as:

$$SSIM(A, B) = \frac{(2\mu_A \mu_B + C_1)(2\delta_{AB} + C_2)}{(\mu_A^2 + \mu_B^2 + C_1)(\delta_A^2 + \delta_B^2 + C_2)} \tag{18}$$

In Equation (18), $\mu_A$ and $\mu_B$ are the average intensity of images A and B, respectively, and $\sigma_A$ and $\sigma_B$ are the standard deviation of images A and B, respectively. $\sigma_{AB}$ is the covariance of A and B, and $C_1$ and $C_2$ are the constants of images A and B, respectively. The higher the SSIM value, the more similar the two images are. The performance of

image fusion can be evaluated by the total amount of similarity between the following fused images F and the source images A and B:

$$M_F(A, B) = SSIM(F, A) + SSIM(F, B) \tag{19}$$

- Mutual information (MI)

  Mutual information [77] is an index to measure the similarity of gray distribution between images A and B, and the probability of gray distribution can be obtained from the image histogram. MI is defined as:

$$I(A, B) = \sum_{a,b} P_{AB}(a, b) \log \frac{P_{AB}(a, b)}{P_A(a) P_B(b)} \tag{20}$$

  In Equation (20), $P_{AB}(a, b)$ represents the joint distribution probability, and $P_A(a)$ and $P_B(b)$ are the distribution probabilities of a and b, respectively.

- Visual information fidelity (VIF)

  An index that is unified in the human visual system and used to assess the efficacy of picture fusion is the fidelity of visual information. Han [78] et al. (2013) used the VIF model to extract the visual data from the source photos, eliminate the distorted data, and eventually accomplish successful visual information fusion. A VIF created specifically for fusion evaluation emerges from the study's integration of all the visual data. Building a model to estimate the distortion between the fused image and the source image is the goal of VIF. There are four steps to this process. The fused image and the source image are first filtered and separated into various blocks. Second, look for distortions in each block's visual data. Third, the fidelity of the visual information in each block is calculated. Fourth, calculate the overall index based on VIF.

**Table 3.** Evaluation effect of image fusion performance index.

| Evaluation Index | Desired Value | Evaluation Effect | Reference |
|---|---|---|---|
| PSNR | Higher value | The higher the proximity, the better the fusion quality | [72] |
| EN | Higher value | The richer the information, the better the fusion quality | [73] |
| MSE | Lower value | The smaller the differences, the better the fusion quality | [74] |
| RMSE | Lower value | The smaller the differences, the better the fusion quality | [75] |
| NCC | Between 0 and 1, ideal value for perfect match is 1 | The better the quality of fusion | [76] |
| SSIM | Between 0 and 1, ideal value for perfect match is 1 | The higher the similarity, the better the fusion quality | [74] |
| MI | Higher value | The more source image information is retained, the better the fusion quality | [77] |
| SD | Higher value | The higher the contrast, the better the fusion quality | [79] |
| AG | Higher value | The richer the information, the better the fusion quality | [80] |

**Table 3.** *Cont.*

| Evaluation Index | Desired Value | Evaluation Effect | Reference |
|---|---|---|---|
| SF | Higher value | The richer the edge texture information, the better the fusion quality | [81] |
| CC | Higher value | The better the quality of fusion | [82] |
| COSIN | Higher value | The more similar the pixels in the image | [83] |

- Standard deviation (SD)
  Standard deviation [79] is based on a statistical concept that reflects the distribution and contrast of a fused image and is used to measure the change in pixel intensity in a fused image. SD is defined as:

$$SD = \sqrt{\sum_{i=1}^{M}\sum_{j=1}^{N}(F(i,j) - \mu)^2} \tag{21}$$

  The average value of the fused image is represented by the variable $\mu$ in Equation (21). Because the human visual system is more sensitive to contrast, high-contrast areas are typically easier to spot, and fusion images will also provide stronger visual effects. High-contrast images typically have bigger SD standard deviations.
- Average gradient (AG)
  The average gradient [80] is used to represent the expressiveness of the texture and detail of the fused image and is usually used to evaluate the sharpness of the image. AG is defined as:

$$AG = \frac{1}{M*N}\sum_{i=1}^{M}\sum_{j=1}^{N}\sqrt{\frac{1}{2}((F(i,j) - F(i+1,j))^2 + (F(i,j) - F(i,j+1))^2)} \tag{22}$$

  In Formula (22), the size of the fused image is $M*N$, and $F(i,j)$ represents the gray pixel of the image at pixel level $(i,j)$. If the average gradient is larger, the fusion image effect is better and the information is richer.
- Spatial frequency (SF)
  Spatial frequency [81] reflects the detailed texture and clarity of the image through the image gradient. The spatial frequency SF can be divided into the spatial row frequency RF and the spatial column frequency CF. SF is defined as:

$$SF = \sqrt{RF2 + CF2} \tag{23}$$

  In Equation (23), the RF and CF distributions are defined as:

$$RF = \sqrt{\frac{1}{M*N}\sum_{i=1}^{M}\sum_{j=1}^{N}[P(i,j) - P(i,j-1)]^2} \tag{24}$$

$$CF = \sqrt{\frac{1}{M*N}\sum_{i=2}^{M}\sum_{j=1}^{N}[P(i,j) - P(i-1,j)]^2} \tag{25}$$

  In Equations (24) and (25), the size of the fused image is $M*N$. According to the above formula, it can be concluded that if you want to calculate SF, you should first calculate the horizontal frequency RF and the vertical frequency CF. The larger the SF value, the richer the edge texture information of the image.
- Correlation coefficient (CC)

The correlation coefficient [82] is an index to measure the degree of linear correlation between the original image and the fused image. CC is defined as:

$$CC = \frac{(r_{I,F} + r_{V,F})}{2} \tag{26}$$

Among them,

$$r_{X,F} = \frac{\sum\limits_{i+1}^{H} \sum\limits_{j=1}^{W} (X(i,j) - \overline{X})(F(i,j) - \overline{F})}{\sqrt{\sum\limits_{i+1}^{H} \sum\limits_{j=1}^{W} (X(i,j) - \overline{X})^2 (\sum\limits_{i+1}^{H} \sum\limits_{j=1}^{W} (F(i,j) - \overline{F})^2)}} \tag{27}$$

In Equation (27), $X$ represents an infrared image (IR) or visible image (VIS). $\overline{X}$ and $\overline{F}$ represent the average pixel values of the source image and the fused image $F$, and $H$ and $W$ represent the length and width of the test image. The larger the CC, the better the fusion performance.

- Similarity (COSIN)

The function of similarity [83] is to convert the corresponding image into a vector and then calculate the cosine similarity of the vector to date. COSIN is defined as:

$$COSIN = \cos(\theta) = \frac{A * B}{\|A\|\|B\|} = \frac{\sum\limits_{i=1}^{n} A_i * B_i}{\sqrt{\sum\limits_{i=1}^{n} (A_i)^2} * \sqrt{\sum\limits_{i=1}^{n} (B_i)^2}} \tag{28}$$

This method requires a lot of calculation, but the result is more accurate. The larger the value of COSIN, the more similar the pixels in the image are.

## 4. Infrared and Visible Image Fusion Dataset

The infrared and visible image fusion datasets, which typically include both infrared and visible image pairs and the two images recorded in different modes (infrared and visible) of the same scene, are a crucial component in the training and testing process of infrared and visible image fusion. The TNO dataset, the RoadScene dataset, the MSRS dataset, etc. are examples of common visible and infrared image fusion datasets. The fundamental elements of each dataset are introduced in this section. The basic elements of each dataset are displayed in Table 4.

**Table 4.** Infrared and visible image fusion datasets.

| Datasets | Basic Content |
|----------|---------------|
| TNO | Contains enhanced vision, near infrared and long wave infrared night images for different military and surveillance scenarios |
| RoadScene | Contains 221 image pairs of highly repetitive scenes |
| MS-COCO | Contains photos of 91 object types, 82 of which have more than 5000 tagged instances |
| MSRS | Contains 1444 pairs of high-quality aligned infrared and visible light images |
| LLVIP | Contains 15,488 pairs of images that are strictly aligned in time and space, mostly taken in dark settings |
| RGBT210 | Contains 210 video sets that are strictly aligned at the pixel level |
| RGBT234 | Contains 234 video sets, expanding the diversity of the scene |
| OTCBVS | Contains 7 infrared datasets, 1 visible light dataset, and 6 visible-infrared datasets |
| INO | Contains 10 different categories of images, a total of 2000, the size of $256 \times 256$ pixels |
| LITIV | Contains a variety of different sensor data and multiple subdata sets |
| GTOT | Contains 50 videos of different scenarios, such as LABS, pools and roads |

### 4.1. TNO Dataset

The TNO dataset [84] is used to create fuzzy target detection algorithms, multi-spectral target detection and recognition algorithms, static and dynamic picture fusion methods, and color fusion algorithms. There are three different image sets that make up the dataset at the moment: the TNO image fusion dataset, the Kayak image fusion sequence, and the TRICLOBS dynamic multi-band image dataset.

The TNO image fusion collection includes nighttime images in the enhanced vision (390–700 nm), near-infrared (700–1000 nm), and long-wave infrared (8–12 m) wavelengths from various military and surveillance scenarios, such as vehicle targets in rural settings. Three incoming kayaks are seen in the Kayak Image Fusion sequence, which combines registered visual, near-infrared, and long-wave infrared image sequences. The recorded visible (400–700 nm), near-infrared (700–1000 nm), and long-wave infrared (LWIR, 8–14 m) motion sequences are included in the TRICLOBS dynamic multi-band picture dataset for the dynamic monitoring of urban environments. Several photos from the TNO dataset are displayed in Figure 10.

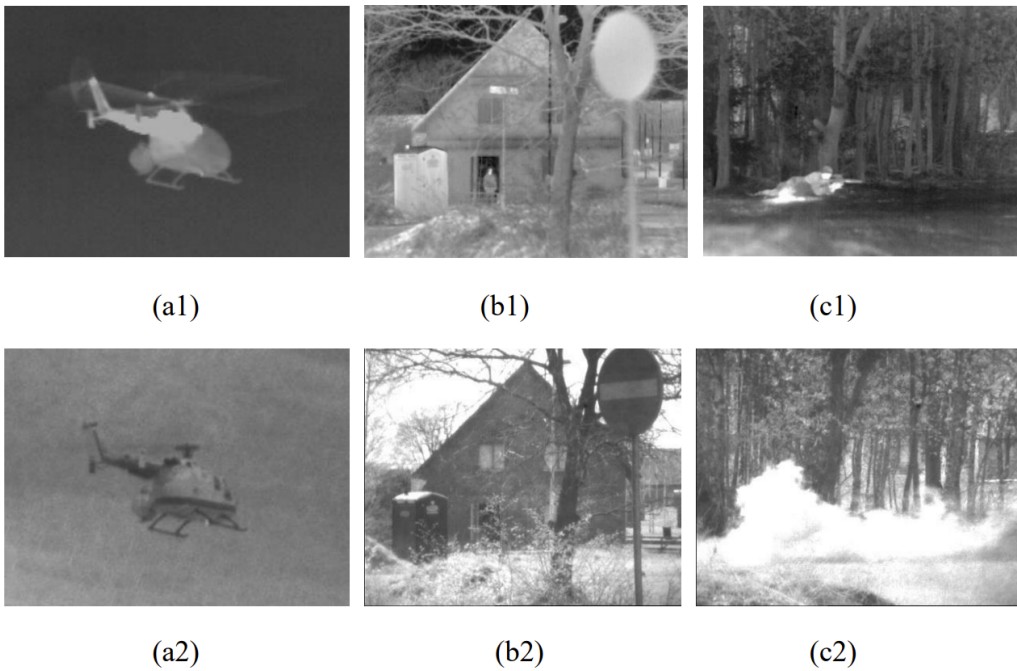

**Figure 10.** Scene images of planes, villages, and forests. (**a1–c1**) infrared images; (**a2–c2**) visible light images.

### 4.2. Roadscene Dataset

The RoadScene dataset is a newly aligned infrared and visible image dataset published by Xu et al. [85] The original infrared image's thermal noise is suppressed after the dataset first chooses image pairs from videos that have a lot of repeating situations. Following meticulous feature point selection, homologous and bicubic interpolation pairs are used to align the picture pairings. The precise registration area is then clipped. Bicubic interpolation is an interpolation algorithm used to forecast continuous pixel values on a discrete pixel grid. At each pixel point in the pixel grid, it is done by utilizing a weighted average of the adjacent pixels. The bicubic interpolation algorithm uses a 16-pixel window and determines the value of the target pixel by taking the weighted average of the 16 pixels around it, taking into account its location in the window. This calculation process is based on an interpolation algorithm for a cubic function, so it is called bicubic interpolation. Bicubic interpolation was selected because it offers improved smoothness and image quality, particularly when aligned across image pairs, minimizing distortion and artifacts between images. For accurate feature matching and the fusion of infrared and visible light

images, this is crucial. The image pairs from the RoadScene dataset that have been aligned offer tremendous convenience for the next studies.

The RoadScene dataset includes 221 pairs of photos with detailed scenarios, including busy roadways, people walking, and moving vehicles. It fixes the issues of the reference dataset's small number of image pairings, low spatial resolution, and lack of detailed information in infrared images. Several pictures from the RoadScene collection are displayed in Figure 11.

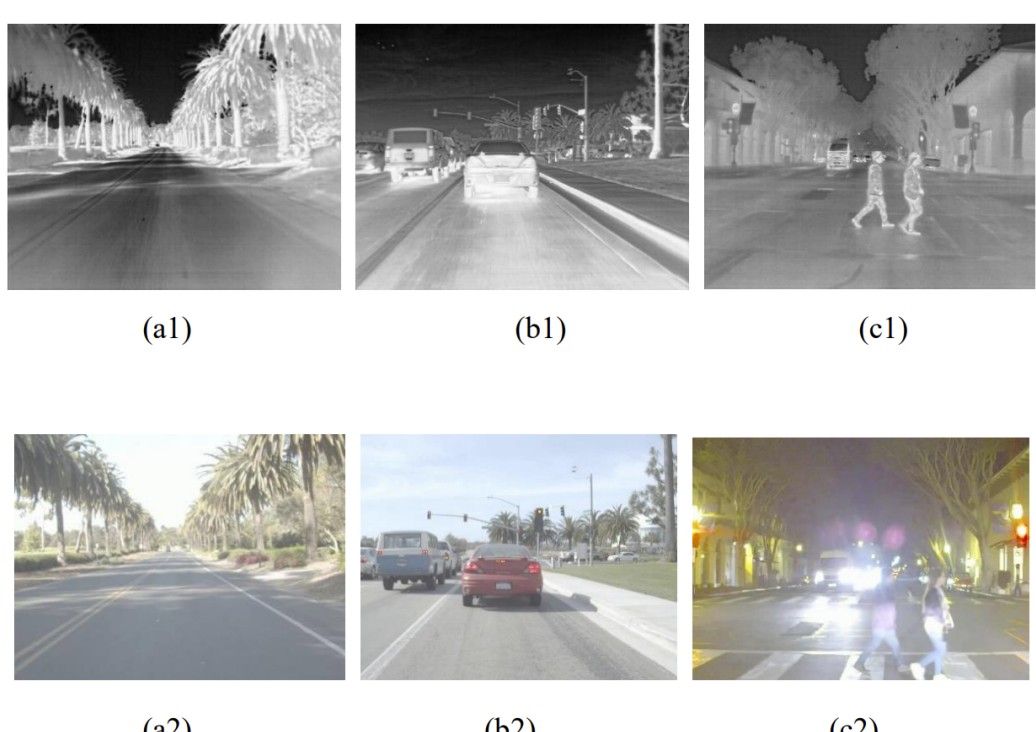

(a1)  (b1)  (c1)

(a2)  (b2)  (c2)

**Figure 11.** Scene images of highway, vehicle and pedestrian. (**a1**–**c1**) infrared images; (**a2**–**c2**) visible light images.

### 4.3. MS-COCO Dataset

The MS-COCO dataset [86] includes images of 91 different item kinds, with more than 5000 labeled instances for 82 of them. A brand-new user interface for category detection, instance localization, and instance segmentation is used to construct the dataset. The performance of fused pictures can be improved by using the grayscale images of the MS-COCO dataset to train deep learning models since the training data for infrared and visible images are insufficient. Several pictures from the MS-COCO dataset are displayed in Figure 12.

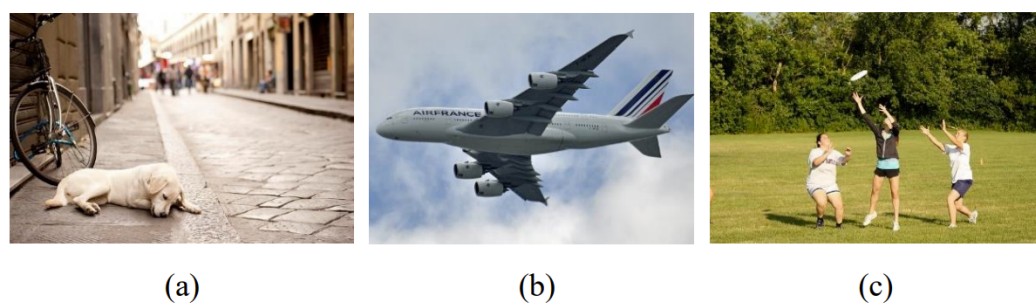

(a)  (b)  (c)

**Figure 12.** Sample images from the MS-COCO dataset. (**a**) images of cities; (**b**) images of aircraft; (**c**) images of lawns

### 4.4. MSRS Dataset

The MSRS dataset, a novel multispectral dataset of infrared and visible picture fusion built on the foundation of the MFNet dataset, has 1444 pairs of highly aligned infrared and visible image pairs. After removing 125 pairs of unaligned image pairs from the MFNet dataset, Tang et al. first collected 715 pairs of daytime images and 729 pairs of nighttime images. Next, they optimized the contrast and signal-to-noise ratio of the infrared images by using a dark channel-based a priori image enhancement algorithm. A previous algorithm based on dark channels was first proposed by He [87] et al. According to this theory, for any image J, its dark channel can be expressed as:

$$J^{dark}(x) = \min_{y \in \Gamma_{(x)}} (\min_{\tau \in \{r,g,b\}} \cdot J^{\tau}(y)), J^{dark} \to 0 \tag{29}$$

where $J^{dark}$ denotes the original image's dark channel J, $\Gamma$ is the three-channel RGB color space, and r is the local area centered on (x, y). From the dark channel, a rough estimate of transmittance can be made:

$$\tilde{t}(x) = 1 - \alpha \min_{y \in \Gamma_{(x)}} \left\{ \min_{\tau} \left[ \frac{I^{\tau}(x)}{A^{\tau}} \right] \right\} \tag{30}$$

where $\alpha$ is the regulator of image fidelity. The final image is:

$$J(x) = \frac{I(x) - A}{\max(t(x) \cdot t_0)} + A \tag{31}$$

where $t_0$ represents the lower limit of transmittance set to avoid the inclusion of noise in the final processing result, usually 0.1. The MSRS dataset's image quality has been significantly enhanced by this technique, and it has eventually given birth to the MSRS dataset. In Figure 13, a number of pictures from the MSRS dataset are displayed.

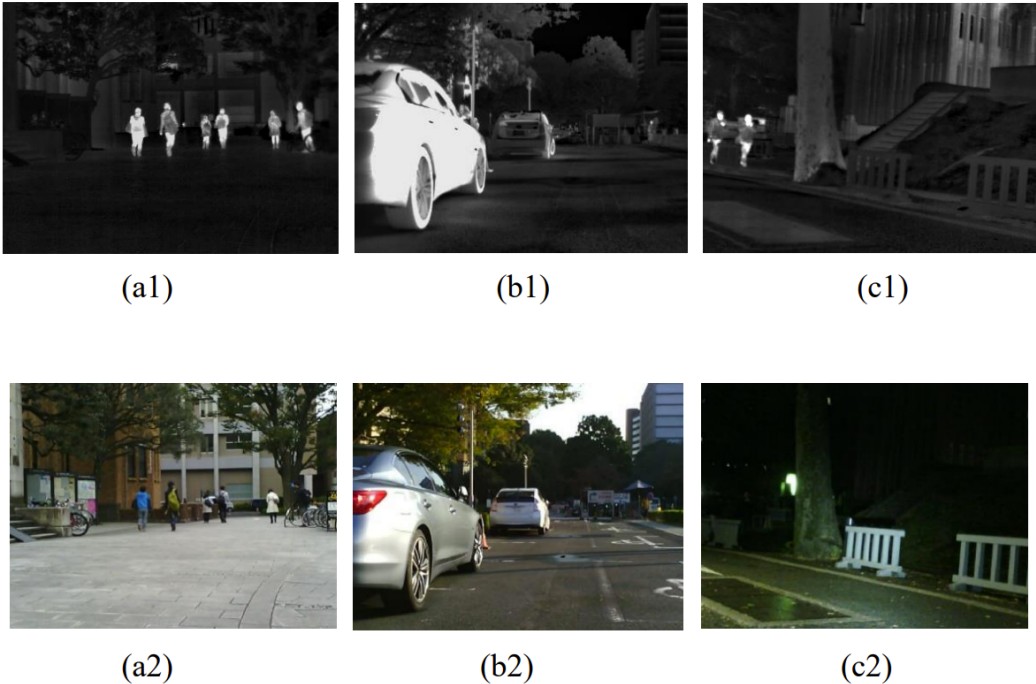

(a1)  (b1)  (c1)

(a2)  (b2)  (c2)

**Figure 13.** Example of MSRS dataset. (**a1**–**c1**) infrared images; (**a2**–**c2**) visible light images.

### 4.5. LLVIP Dataset

The LLVIP dataset [88] is a low-light vision-infrared pairing dataset. The collection contains 30,976 photos and 15,488 pairs of photographs, the majority of which were captured in very dark situations, and all images are strictly time- and space-aligned. Additionally, a substantial variety of pedestrian photos taken in poor light are included in the dataset. Low-light visible photos are given rich supplemental information by infrared photographs. Furthermore, this dataset has unusually good image quality; the original visible light image's resolution is 1920 × 1080, while the infrared image's resolution is 1280 × 720. These benefits all point to the LLVIP dataset's high value in the area of detecting pedestrians in low light. The LLVIP dataset will help to advance the field of computer vision by enabling the use of image fusion, pedestrian recognition, and image-to-image conversion in low-light vision. Figure 14 depicts a selection of photos from the LLVIP dataset.

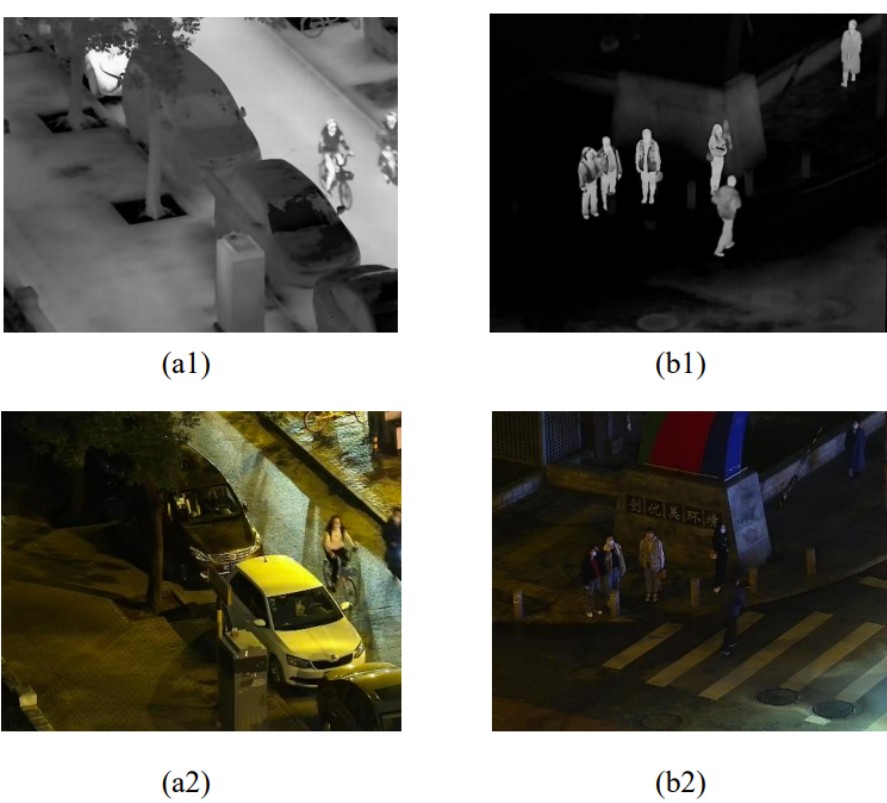

(a1)　　　　　　　　　　　　　　　(b1)

(a2)　　　　　　　　　　　　　　　(b2)

**Figure 14.** LLVIP dataset diagram. (**a1**,**b1**) infrared images; (**a2**,**b2**) visible light images.

### 4.6. Datasets of RGBT210 and RGBT234

The RGBT210 dataset [89] was captured using a thermal infrared imager (DLS-H37 DM-A) and a CCD camera (SONY EXView HAD CC). The two cameras have identical image parameters, and the optical axis is parallel across the collimator so that the two cameras' common horizontal lines are aligned at the pixel level. The collection contains 210 video sets totaling around 210,000 frames. Figure 15 depicts a selection of photos from the RGBT210 dataset.

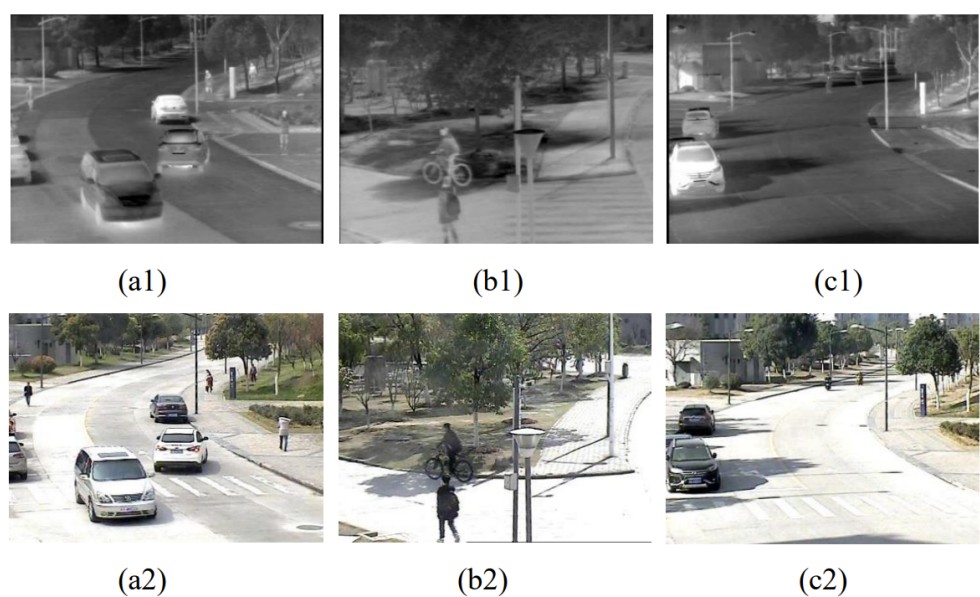

**Figure 15.** Example of RGBT210 dataset. (**a1–c1**) infrared images; (**a2–c2**) visible light images.

In comparison to the RGBT210 dataset, the RGBT234 dataset [90] enhances the diversity of sceneries and adds movies shot in hot weather, including 234 video sets, approximately 233,800 frames, and each video set includes the visible and infrared video sequences of the video. Figure 16 depicts a selection of photos from the RGBT234 dataset.

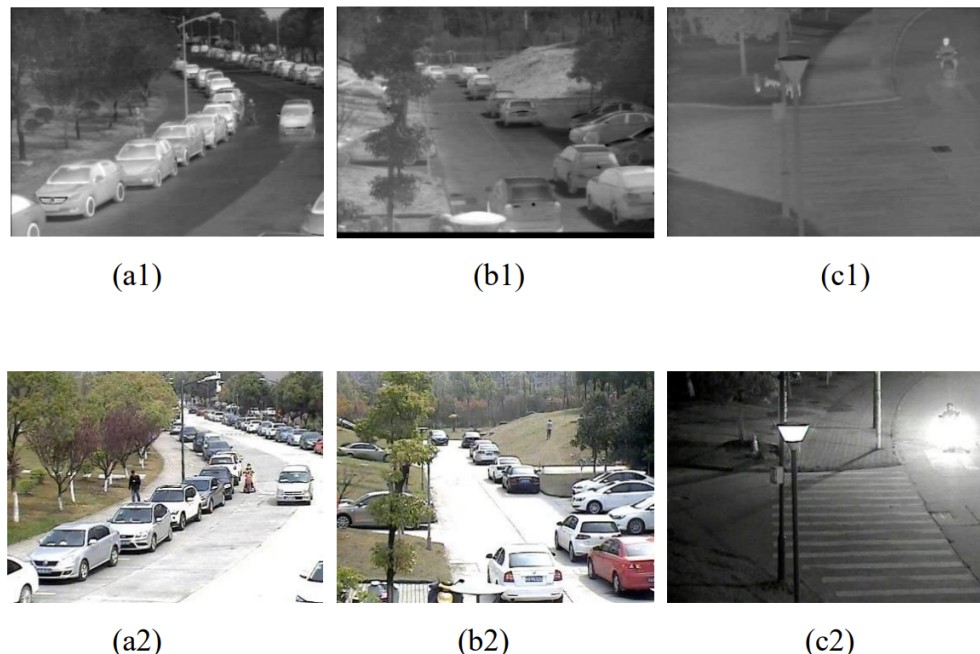

**Figure 16.** Example of RGBT234 dataset.(**a1–c1**) infrared images; (**a2–c2**) visible light images.

### 4.7. Other Datasets

In addition to the above datasets, there are also some public datasets, such as the OTCBVS dataset, the INO dataset, the LITIV dataset, the GTOT dataset, and the VOT-2016 dataset.

The OTCBVS dataset [91] is a freely accessible benchmark dataset for testing and evaluating algorithms in the visible and infrared domains. The dataset is divided into 14 sub-datasets, comprising seven infrared datasets, one visible light dataset, and six

vision-infrared datasets, which include pedestrians, faces, motions, weapons, vehicles, and ships.

The INO dataset has ten image categories, each with 200 images, for a total of 2000 images with a resolution of 256 × 256 pixels. This dataset can be used to train and test image recognition algorithms. Each image category symbolizes a different object or scene, such as a cat, dog, automobile, or airplane.

The LITIV dataset [92] is a multi-modal dataset for computer vision and machine learning research. The collection includes a wide range of diverse sensor data, including photos, movies, infrared photos, depth photos, etc. There are numerous sub-datasets in it as well. A dataset for target detection and tracking, which includes a number of different target types and scenarios, is one of the sub-datasets.

The GTOT dataset [93] contains about 15,800 frames of video in 50 different scenarios, such as laboratories, pools, and roads, with objects such as swans and vehicles.

## 5. Application of Infrared and Visible Image Fusion

### 5.1. Application of Night Vision

The target object's or scene's thermal radiation information is typically converted into false-color images because the human visual system is more sensitive to color images than to grayscale photos. Due to the use of color transfer technology, the resulting color image has a realistic daytime color appearance, which makes the scene more intuitive and aids the spectator in understanding the image. Figure 17 illustrates an instance of fusing visible and infrared images for color vision at night.

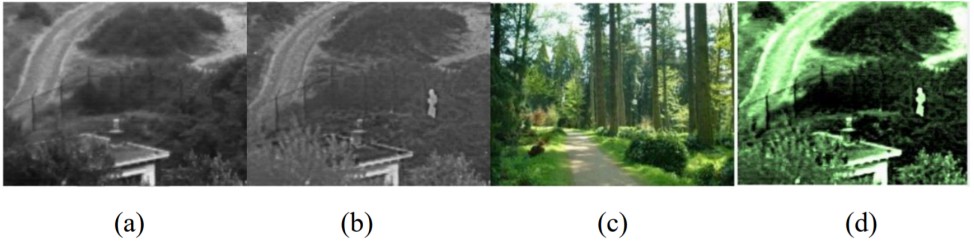

(a)       (b)       (c)       (d)

**Figure 17.** Example of infrared and visible image fusion for color vision at night. (**a**) visible light image; (**b**) infrared image; (**c**) Reference image; (**d**) Fused image.

Grayscale images are less responsive to human vision than color images. Human eyes are capable of distinguishing thousands of colors, but they are only capable of doing so for about 100 grayscale images. For this reason, it is imperative to colorize grayscale images, especially since the fusion method of infrared and visible images with color contrast enhancement has been widely adopted in military equipment [94]. Additionally, there is now more interest in the color fusion ergonomics representation of many image sensor signals due to the quick growth of multi-band infrared and night vision systems.

### 5.2. Applications in the Field of Biometrics

The topic of facial recognition research has advanced quickly. The face recognition technology for visual images has advanced to a very advanced stage and has had great success [95]. The rate of face recognition using the visual technique will decrease in low-light situations; however, thermal infrared face recognition technology can perform well. Figure 18 shows a face image captured in both infrared and visible light.

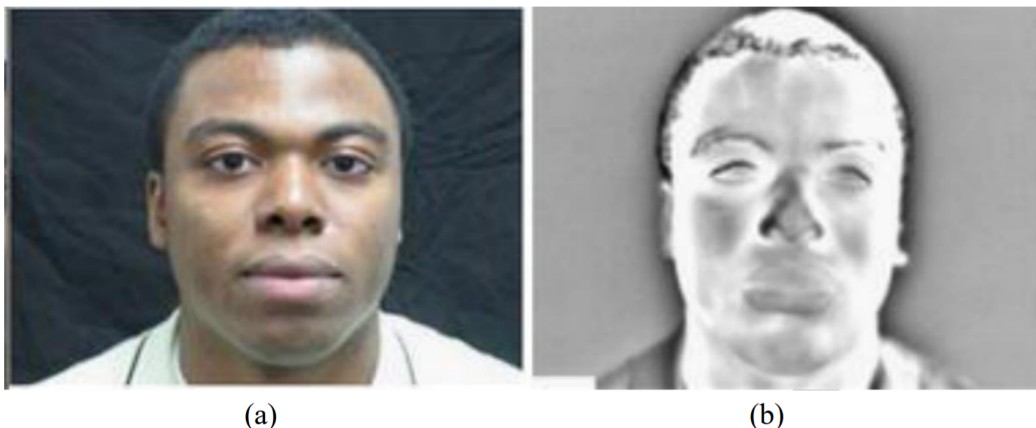

$$(a) \qquad\qquad\qquad (b)$$

**Figure 18.** Example of infrared and visible face images. (**a**) visible light image; (**b**) infrared image.

Although visible image-based face recognition technology has been thoroughly researched, there are still significant issues with its practical implementation. For instance, the recognition effect will be significantly impacted by the illumination in the actual scenario, changes in facial expression, background, etc. For the purpose of recognizing faces, infrared photographs can supplement the information that is hidden in visible photos. Recent years have seen a rise in the application of infrared and visible light picture fusion based on biometric optimization algorithms. By increasing the quantity of computing, this kind of approach can increase recognition accuracy and give biometrics more supplementary data. The use of infrared and visible light fusion technologies in the field of biometrics will also become more widespread in the future.

The growing use of facial recognition technology, however, also brings up several ethical and privacy concerns. For instance, while surveillance systems in public spaces contribute to society's safety, they also raise questions about whether people's facial information might be stolen and used inappropriately by outside parties. Concerns regarding abuses and violations of human rights have been raised in various nations where governments are utilizing facial recognition technology to monitor citizens' activities. As a result, we must develop laws, rules, and moral standards to control the use of facial recognition technology and stop abuse. To increase the accuracy and fairness of technology, we also need to put in place an appropriate regulatory framework. Only in doing so will we be able to more fairly weigh the benefits and drawbacks of facial recognition technology and, in the end, establish the peaceful coexistence of science, technology, and human values [96].

### 5.3. Application in Detection and Tracking Field

In the area of target detection, visible and infrared pictures can work in synergy to detect targets. Bulanon [32] et al. merged the thermal picture and the visual image of the orange tree canopy scene to overcome the limitations of the two imaging techniques and increase the accuracy of fruit detection. First, the fruit can be seen in the visible picture due to the color difference between the fruit and the tree crown, but since the visible image is sensitive to light fluctuations, the fruit may be misclassified. Following the real test, it is clear that the temperature of the fruit in the evening is obviously greater than that of the tree top, allowing the produced infrared image to effectively detect the fruit. Finally, the precision of fruit detection has improved. For the purpose of enhancing the overall effectiveness of the monitoring system, Elguebaly et al. [97] proposed a target detection method based on the fusion of visible and infrared images.

In succeeding frames of the video, object tracking locates the object specified in the current frame. In order to locate the target item in a time series, the target tracking algorithm must ascertain the relationship between the frames. Single-mode tracking is the most popular but is less robust. Target tracking performance cannot be assured if it is nighttime or there are poor lighting conditions because the quality of the visible picture is

highly tied to the imaging environment. This is similar to how infrared photographs lack texture, have a poor understanding of the scene's three dimensions, and cannot guarantee their performance in specific situations. Therefore, Liu [98] et al. proposed a visual tracking method that integrates color images and infrared images, namely, RGBT tracking, which can fuse complementary information in infrared and visible images to make target tracking more robust.

*5.4. Applications in the Field of Medical Diagnosis*

Medical image fusion seeks to enhance image quality by maintaining particular features, expanding the use of images in clinical diagnostics, and assessing medical issues. Medical image fusion has increasingly proven to provide essential benefits as clinical application requirements continue to advance. Through the study of medical images, computers or clinicians are able to make the majority of medical diagnoses. Different types of medical images employ various imaging techniques and place various emphases on the description of the human body. Computed tomography [99] (CT), magnetic resonance imaging [100] (MRI), single photon emission computed tomography [101] (SPECT), positron emission tomography [102] (PET), and ultrasound [103] are examples of common medical techniques. These methods include those that concentrate on regional metabolic power and those that concentrate on organ structure. The effectiveness and precision of diagnosis will be significantly increased, while redundant information will be removed and picture quality will be raised if medical images from various modes can be merged. As seen in Figure 19, MRI pictures can better see soft tissues that are more deformed, whereas CT scans can better observe denser and less distorted tissues. The two different types of photos can then be combined using image fusion technology to deliver more precise patient information.

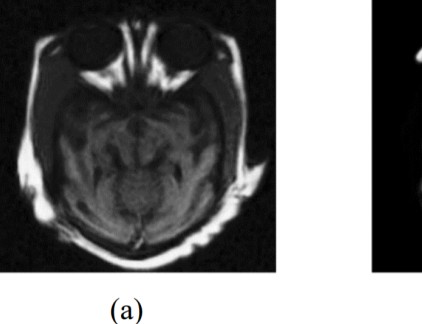 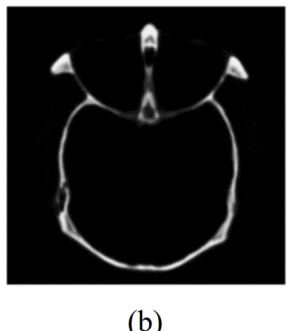 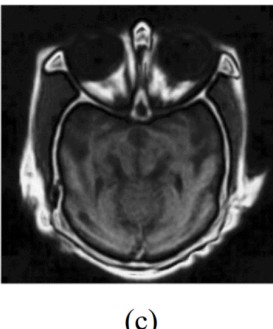

(a)          (b)          (c)

**Figure 19.** Image fusion in clinical imaging field. (**a**) MRI; (**b**) CT; (**c**) Fused image.

*5.5. Applications in the Field of Autonomous Vehicles*

The field of autonomous vehicles is constantly evolving, and vision, radar, and LiDAR sensors are also widely used in autonomous vehicle perception technology. The first step in autonomous driving and a crucial element in deciding how well a vehicle performs is the ability to precisely and properly recognize the surroundings around it. General vision sensors may have trouble identifying things, particularly in low light, intense sunlight, or severe weather, which is a test for self-driving automobiles. In order to increase the visual effects of automated vehicles and the safety and dependability of autonomous driving in challenging conditions, infrared and visible light image fusion has been introduced into the application of the field of driverless vehicles. For usage in the field of autonomous cars, Li [104] et al. suggested a novel two-stage network (SOSMaskFuse) that can efficiently cut down on noise, extract critical thermal data from infrared photos, and display more texture features in visible images. In order to efficiently detect and identify objects even in environments with limited visibility, such as day or night, Choi [105] et al. presented a sensor fusion system that integrates thermal infrared cameras with LiDAR sensors. This system effectively ensures the safety of autonomous vehicles.

## 6. Future Prospects

Infrared and visible picture fusion, which has moved from conventional algorithms to deep learning algorithms, has produced positive results as a result of the ongoing development of information fusion. Numerous image fusion problems have significantly benefited from the introduction of deep learning technologies. Recent years have seen researchers in this area continually suggest new network architectures and training methods to further the idea of deep learning. For instance, Tang [27] et al. presented the DIVFusion framework, which combines dual-mode fusion and low-light enhancement to produce information fusion images that are more effective and increase visual tasks. To more clearly express the texture details in the fused image, Li [106] et al. proposed a method for merging infrared and visible images using residual dense networks and gradient loss. The proposed gradient loss network can be well matched with the special weight blocks extracted from the input image to express the details in the image more clearly. These are all illustrations of deep learning architectures that have shown promising results, demonstrating the enormous potential of deep learning. However, infrared and visible light fusion using deep learning still confronts certain difficulties:

(1) Multi-modal image registration
Infrared and visible images are typically distinguished by distinct projections, resolutions, and geometric distortions during the imaging process because they are subject to different physical laws and imaging properties at the moment of acquisition. For instance, this misalignment between IR and visible light significantly lowers the performance for practical applications in the field of pedestrian detection [107]. Both the current RGBT234 and M3FD datasets have this issue. To create spatial and temporal consistency, it is crucial to figure out how to tackle the difficulties of geometric distortion, rotation, scale, and viewpoint differences between them. The following are some existing methods to achieve multi-modal image registration: Ye [108] et al. proposed a robust matching method (SFOC) based on controllable filters, the core idea of which is to establish a fast similarity measurement method using FFT technology and integral images. It has excellent performance in registration accuracy and computational efficiency but lacks scale and rotation invariance. Yao [109] et al. proposed a robust NMR matching method based on co-occurring filtering (CoF) spatial matching (CoFSM). Its advantage is that the NRD difference problem can be transformed into an optimization problem using image feature similarity information, but the computational complexity is high. From this point of view, each method has certain disadvantages, so it is extremely challenging to perfectly align infrared and visible light images.

(2) Dynamic range fusion
Between infrared and visible images, there are variances in the dynamic range. Although infrared images often have a wide dynamic range, they are devoid of fine details. Visible images offer more resolution, but they have a constrained dynamic range. Regarding the technology of high dynamic range fusion, Kang [110] et al. suggested a method to execute high dynamic range fusion on video data that automatically collects two images with various exposures and achieves real-time image collection using high dynamic range fusion technology. After a challenging offline registration process, the technique combines the two inputs with the weighted average to provide a high-dynamic-range image. The challenge of automatically creating high-dynamic-range panoramic images from input photographs with significant geometric and photometric variation was overcome by Eden [111] et al. To decide which input photographs contain the most useful pixel information, they employ a graph-cutting technique. The input image's pixels are reassembled to create the final image. These two techniques can produce high-dynamic-range images from a small number of input photographs, but they are unable to provide additional image enhancement. There is still an issue to be solved on how to maintain the balance of detailed informa-

tion and contrast in the fusion process while maintaining the dynamic range of the two images.

(3) Real-time image fusion
Real-time performance is a crucial need in various application scenarios, including surveillance, night vision, autonomous vehicles, and the military and security sectors. The development of infrared and visible image fusion in real-world applications will be hampered by the existing infrared and visible image fusion algorithms' potential performance limitations when processing large-scale images, which takes a lot of time and computational resources. Sedat et al. attempted to create a deeper learning-based approach to infrared and visible picture fusion in literature [112]; however, there is currently very little study in this area. The challenge remains in figuring out how to further reduce computer complexity, lower memory, and processor needs while maintaining good fusion outcomes and accomplishing high-quality real-time fusion with constrained computing resources.

(4) Adaptability to specific scenarios
For infrared and visible picture fusion, different application scenarios have varied conditions and requirements. There is currently no general algorithm that can be used in all situations. It is occasionally required to undertake research for a particular environment in order to suit the needs of unique application scenarios. The majority of the visible and infrared image fusion techniques now in use do not consider their practical use. Many redundant or even erroneous pieces of information are added during the fusion process as a result of some end-to-end models' failure to discriminate between different areas of various source images while building loss functions, which weakens the valuable information in the fusion image. For instance, the literature in [25] provides a typical example. U2Fusion is a typical method based on CNN, and FusionGAN is a typical method based on GAN. When these two approaches are used for object detection, it turns out that FusionGAN decreases the background texture, whereas U2Fusion weakens the prominent object. Both approaches fail to fully exploit their advantages. In order to achieve the detection of salient targets as well as the extraction and reconstruction of useful information, Ma et al. presented the STDFusionNet technique. Rich textures can still be seen while highlighting significant targets in the merged photos. The techniques for fusing infrared and visible images are currently becoming more and more varied, and the complexity of the scenes to be handled is rising. It is a very challenging task to develop an algorithm that can adapt to various scenarios and perform well on various indicators. As a result, we must commit to conducting additional research in order to develop more flexible and reliable picture fusion techniques and offer the best solutions for certain scenes.

(5) Standard, comprehensive integration evaluation indicators
There is no established fusion evaluation index in the field of image fusion, and the fusion findings frequently perform well for some indications but poorly for others [113]. As a result, it is impossible to evaluate the performance of the picture fusion method accurately and properly. Therefore, in order to evaluate the image fusion algorithm thoroughly, it is important to offer a fairly full fusion evaluation index.

(6) Converter-based infrared and visible image fusion method
Converter-based techniques for performing infrared and visible picture fusion were introduced in the field of image fusion in 2021. The converter in this method converts the visible and infrared images into a particular feature space, and the features into two features. The fusion technique fuses the spaces, and then the fused feature space is transformed into the image space to produce the fused image. An infrared and visible image fusion technique based on a converter and generative adversarial network was proposed by Rao [114] et al. The converter fusion module in the generator is made up of the spatial converter and the channel converter merged. The transformer is also used in other phases of the VIF approach, among other approaches. For example, to fuse the features of infrared and visible images together, Yang [115] et al. suggested

the YDTR approach, which integrates CNN and transformer in the encoding and decoding branches. There are various benefits to using converter-based techniques for fusing images from the visible and infrared spectrums. For instance, the converter can more quickly learn the mapping relationship between the two separate picture fusion modes. Additionally, the algorithm is adaptable, and the converter's settings can be changed by various tasks to enhance the quality of the fusion. However, there are few infrared and visible image fusion methods based solely on converters, which are all applied to fusion tasks by combining converters with other networks. Transformer-based techniques are still in their infancy as an application in the fusion of infrared and visible images, but they will undoubtedly advance in the next few years.

## 7. Conclusions

This study provides a brief overview of infrared and visible picture fusion technology as well as an understanding of its historical evolution and background characteristics. The most crucial step is to arrange infrared and visible image fusion techniques simply, including compressed sensing, sub-space, mixed, and other techniques. The main trend in infrared and visible picture fusion in the future is unquestionably deep learning. Then, several infrared and visible image fusion objective assessment indexes are highlighted, and particular calculation formulas are introduced. The datasets for visible and infrared images are then quickly sorted and introduced. Additionally, the infrared and visible image fusion application sectors are also introduced, including night vision, medical diagnostics, biometrics, detection, and tracking. At this time, infrared and visible image fusion technology is advanced, fusion quality is increasing, and new infrared and visible image fusion techniques are continually being developed. Future advancements in infrared and visible image fusion will also be made in uncharted territories, advancing human knowledge and technology.

**Author Contributions:** Y.L.: conceptualization, methodology, software, validation, formal analysis,investigation, resources, data curation, writing—original draft. Z.L.: Writing—Review, Editing, Supervision, Project administration, Resources. All authors have read and agreed to the published version of the manuscript.

**Funding:** This work was supported in part by the National Natural Science Foundation of China under Grant 61801319, in part by Sichuan Science and Technology Program under Grant 2020JDJQ0061, 2021YFG0099, in part by Innovation Fund of Chinese Universities under Grant 2020HYA04001, in part by Innovation Fund of Engineering Research Center of the Ministry of Education of China, Digital Learning Technology Integration and Application (No. 1221009), in part by the 2022 Graduate Innovation Fund of Sichuan University of Science and Engineering under Grant Y2023297.

**Institutional Review Board Statement:** Not applicable.

**Informed Consent Statement:** Not applicable.

**Data Availability Statement:** Publicly available datasets were analyzed in this study. This data can be found here: TNO: https://figshare.com/articles/dataset/TNO_Image_Fusion_Dataset/1008029; INO: https://www.ino.ca/en/technologies/video-analytics-dataset/videos/; RoadScene: https://git\hub.com/hanna-xu/RoadScene; MSRS: https://github.com/Linfeng-Tang/MSRS; LLVIP: https://bup\t-ai-cz.github.io/LLVIP/.

**Conflicts of Interest:** No potential conflict of interest was reported by the authors.

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
