# Peer review of "Infrared and Visible Image Fusion: Methods, Datasets, Applications, and Prospects"

_applsci, doi:10.3390/app131910891_

Round 1

Reviewer 1 Report

Infrared and visible light image fusion combines infrared and visible light images by extracting key information from each image and fusing them together to provide a more comprehensive image with more features from the two photos. Fusion of infrared and visible images has gained popularity in recent years and is increasingly used in sectors such as target recognition and tracking, night vision, scene segmentation and others. This paper is organized as follows: it is opened by a broad introduction. In the second section, infrared and visible image fusion techniques are explained in depth and the benefits and drawbacks of each technique are highlighted. The third section lists the fusion image evaluation indices and explains how these indices evaluate the quality of the fusion image. The infrared and visible image fusion data sets are sorted in the fourth section and the differences between the different data types are highlighted. The fifth section provides a brief overview of infrared and visible image fusion applications. The sixth section describes the problems and potential future perspectives for infrared and visible image fusion research. In the introduction section, the authors carefully analyze the studied field. An extensive bibliography is supported, going up to the current one, the year 2023. According to the initial research results, the image fusion methods can be roughly divided into the following two categories: the spatial domain fusion method and the transform domain fusion method. In spatial domain fusion, we process the image pixels directly to produce the desired image result. In the transform domain fusion, the image will first be transferred to the frequency domain, we need to perform Fourier transform image processing and then perform inverse Fourier transform to get the image result we want. The original image classification techniques can no longer meet the requirements because of the advances in image fusion technology in recent years. According to the existing theoretical technologies, there are several different types of image fusion techniques that can be classified, including multi-scale decomposition, sparse representation, neural networks, subspace, significance, hybrid, compressed detection and deep learning. Valid and efficient processing methods supported by a well-developed mathematical apparatus are carefully described. Then follows a series of examples through which the analyzed methods are implemented, emphasizing the role of each one in the analysis of the images and the importance in highlighting the details. Then follows a series of future research directions which the authors want to apply in the future to the study. This study provides a brief overview of infrared and visible image fusion technology and an understanding of its historical evolution and background characteristics. The most important step is the simple arrangement of infrared and visible image fusion techniques, including compressed, subspace, mixed and other sensing techniques

Author Response

请参阅附件。

Reviewer 2 Report

This paper “Infrared and Visible Image Fusion: Methods, Datasets, Applications and Prospects” aimed to explore historical context of infrared and visible image fusion: methods, datasets, applications, and prospects. The complete text is summarized, and the current infrared and visible image fusion field is prospected.

The topic is justified. The paper could be further improved if the following remarks are taken into consideration:

1.       ABSTRACT: unfold the abstract, in terms to describe little bit more about the overall infrared and visible image fusion: methods, datasets, applications and prospects. Currently, the overall, text is skewed towards background information only.

2.       A few of the grammatical mistakes were found in the whole draft.

3.       Introduction section lacks justification of the research, however, background etc. is well written.

4.       Rest of the article is well written and organized.

a few of the grammar mistakes.

Reviewer 3 Report

applsci-2605362

Infrared and Visible Image Fusion: Methods, Datasets, Applications and Prospects

Overall, the paper holds promise, but substantial revisions are essential to elevate its quality. By addressing the below points and focusing on creating a coherent, detailed, and engaging narrative, the paper can effectively communicate the complexities and advancements in the field of infrared and visible image fusion.

Comments:

  1. Elaborate on how exactly the process of extracting main information from both infrared and visible images is performed for the fusion process? It would be helpful to provide more insight into the specific benefits that infrared and visible image fusion brings to the various sectors authors mentioned.
  2. Clarify the criteria used to select the common datasets like TNO, RoadScene, MSRS, and LLVIP for this study?
  3. How does the presence of noise or artifacts in the input images affect the quality of the fused image? Are there any denoising techniques integrated into the fusion process?
  4. Regarding the TNO dataset, are there plans to expand the variety of scenarios and scenarios covered in the dataset in the future?
  5. Explain the rationale behind choosing bicubic interpolation for aligning image pairs in the RoadScene dataset? Were other interpolation methods considered and tested?
  6. How do authors see the MS-COCO dataset contributing to the field of infrared and visible image fusion, considering its primary focus on instance segmentation and object detection?
  7. In the MSRS dataset, the enhancement of contrast and signal-to-noise ratio for infrared images is mentioned. Could authors provide more technical details about this enhancement algorithm?
  8. Given the increasing importance of low-light vision in various applications, could authors discuss the potential impact of the LLVIP dataset on advancing research in this area?
  9. Can authors elaborate on how the fusion of infrared and visible images improves the accuracy of target detection in scenarios like the orange tree canopy scene?
  10. The integration of deep learning into infrared and visible image fusion is mentioned. Could authors provide examples of specific deep learning architectures that have shown promising results?
  11. Multi-modal image registration is highlighted as a challenge. What are some existing approaches to address this issue, and how effective have they been so far?
  12. Regarding dynamic range fusion, could authors provide examples of techniques that have been proposed to balance detailed information and contrast in fused images?
  13. Real-time image fusion is a crucial requirement in various fields. Are there any ongoing efforts to develop more efficient algorithms for real-time infrared and visible image fusion?
  14. Can authors provide examples of specific application scenarios in which existing fusion algorithms may not perform optimally due to the unique conditions of those scenarios?
  15. Regarding the lack of established fusion evaluation indices, could authors suggest potential parameters or metrics that could be used to comprehensively evaluate fusion performance?
  16. authors mentioned the YDTR approach as a converter-based technique. Could authors explain the fundamental concept of converter-based fusion methods and their advantages over other approaches?
  17. As deep learning continues to evolve, do authors foresee any specific challenges or limitations that might arise when applying deep learning to infrared and visible image fusion?
  18. How would authors suggest researchers strike a balance between the increasing complexity of fusion methods and the need for practical, real-world applicability?
  19. Are there any plans to create standardized guidelines or benchmarks for evaluating the quality and effectiveness of infrared and visible image fusion techniques?
  20. With the integration of deep learning into fusion, how do authors envision the role of traditional image processing techniques evolving in the context of infrared and visible image fusion?
  21. In terms of future advancements, could authors provide some insights into how infrared and visible image fusion might contribute to emerging technologies like autonomous vehicles?
  22. Could authors expand on the potential challenges and complexities associated with adapting infrared and visible image fusion algorithms to specific and diverse application scenarios?
  23. Considering the rapid pace of technological development, how do authors plan to keep this study's information current and reflective of the latest advancements in the field?
  24. Could authors provide some examples of recent research works or breakthroughs that showcase the promising potential of infrared and visible image fusion?
  25. Given the increasing availability of multi-sensor imaging systems, do authors think the concept of fusion might extend beyond just infrared and visible images in the future?

Minor editing of English language required.

Reviewer 4 Report

The article begins by providing a historical context for infrared and visible image fusion, which is essential for understanding its evolution over time. This background information helps readers grasp the significance of recent developments. It effectively outlines both domestic and international research efforts in the field, giving readers a sense of the global landscape of image fusion. This is valuable for researchers and practitioners looking to stay up-to-date with the latest advancements.

The article categorizes common datasets and analyzes specific application fields of infrared and visible image fusion. It offers practical examples to illustrate the versatility and real-world applicability of this technology, making it accessible to a broad audience. The article also identifies the transition from conventional algorithms to deep learning as a significant advancement in image fusion. It highlights ongoing challenges such as multi-modal image registration and real-time fusion, which helps set research directions for the future. The article acknowledges the need for adaptability to specific scenarios, which is a crucial consideration in real-world applications. This recognition of context-specific challenges adds depth to the discussion.

While all above great points that this article delivers, it also lacks one key aspect such as article does not touch upon the ethical and privacy implications of infrared and visible image fusion, which are becoming increasingly important in fields such as biometrics and surveillance. Please add some information about it if possible. 

Nevertheless, it serves as a useful review for those interested in the topic and seeking a general understanding of its evolution and applications. The future of infrared and visible image fusion is indeed promising, and continued research will likely lead to significant advancements in the field. 
